# When does dough become a bagel?
# Analyzing the remaining mistakes on ImageNet

**Vijay Vasudevan, Benjamin Caine, Raphael Gontijo-Lopes, Sara Fridovich-Keil[2], Rebecca Roelofs**
{vrv, rofls}@google.com
Google Research, Brain Team, [2]University of California, Berkeley

## Abstract

Image classification accuracy on the ImageNet dataset has been a barometer for progress in computer vision over the last decade. Several recent papers have questioned the degree to which the benchmark remains useful to the community [33, 3, 31, 42, 36], yet innovations continue to contribute gains to performance, with today's largest models achieving 90%+ top-1 accuracy. To help contextualize progress on ImageNet and provide a more meaningful evaluation for today's state-of-the-art models, we manually review and categorize every remaining mistake that a few top models make and provide insights into the long-tail of errors on one of the most benchmarked datasets in computer vision. We focus on the multi-label subset evaluation of ImageNet, where today's best models achieve upwards of 97% top-1 accuracy. Our analysis reveals that nearly half of the supposed mistakes are not mistakes at all, and we uncover new valid multi-labels, demonstrating that, without careful review, we are significantly underestimating the performance of these models. On the other hand, we also find that today's best models still make a significant number of mistakes (40%) that are obviously wrong to human reviewers. To calibrate future progress on ImageNet, we provide an updated multi-label evaluation set, and we curate **ImageNet-M**ajor[1]: a 68-example "major error" slice of the obvious mistakes made by today's top models—a slice where models should achieve near perfection, but today are far from doing so.

## 1 Introduction

Computer vision models often evaluate their performance on the ImageNet classification dataset [5, 30] and many variants [29, 12, 13, 11, 38], as a signal of capability for visual understanding. As performance on the standard sets have reached diminishing returns to top-1 and top-5 accuracy, much recent work [33, 3, 29, 31, 36, 17] has focused on understanding what is left for the computer vision community to solve, and where the community should be driving toward. Prior studies of ImageNet errors have identified issues stemming from lack of multi-labels, label noise, under-specified classes, and more [33, 31, 3, 36, 18].

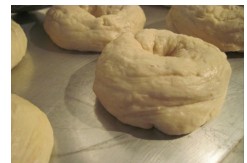

Label: dough; Model: bagel. When does dough become a bagel?

Label errors and label noise affect the evaluation of any model [25], and ImageNet is no exception. Many studies above have spent effort to correct and improve these labels, showing that while ImageNet performance improvements are approaching diminishing returns, the dataset can remain useful to the community, but only if we collectively continue to shepherd it. As the best models improve, however, it is becoming increasingly challenging to assess the often novel predictions these models make. For example, should we penalize models for

---

[1]Dataset and analysis available at `https://github.com/google-research/imagenet-mistakes`.

being the first to predict that a pre-baked bagel may be a bagel, as one of the models we review in this work does?

Machine learning models tend to make mistakes with varying severity and importance (a function of both the prediction as well as the label definitions), and previous studies [33, 3] on ImageNet have shown that non-experts find it challenging to determine the correctness of a model's prediction. Our own experience, highlighted by the doughy-bagel, is that many of the remaining mistakes these top models make are quite reasonable and probably should not be considered mistakes—understanding the severity and type of these remaining mistakes can help calibrate our barometer of progress.

Indeed, to our knowledge, there has not been an expert-review, categorization, and severity assessment of the remaining long-tailed mistakes, which becomes particularly important at these margins. Our experience working with production teams on deployed applications has suggested that manual triage and assessing individual failures provides a useful indicator of model performance that aggregate measures can fail to capture. Thus, in this work we attempt to analyze (as expert reviewers) *every remaining mistake* that a few state-of-the-art models make to better understand (a) which of the remaining mistakes remain egregious errors, (b) what error category they might fall in, and (c) what evaluations might capture the most important long-tail failures that remain.

In this paper we analyze the ImageNet multi-label validation subsets [31], in which expert labelers were used to assess the correctness of model predictions through the year 2020, and on which a 1000-image human-evaluated subset provides a direct comparison to expert human performance. By analyzing the mistakes of two large 2022-era ImageNet models, we found that:

- **Nearly half of each model's mistakes were deemed correct** under a careful, expert multi-label re-evaluation, halving the error rate. Had we not analyzed the models' mistakes, we would be severely underestimating the models' actual performance.

- **Approximately 40% of the remaining mistakes can be classified as 'major' errors**: errors that most humans would likely not make, suggesting that many of the long-tailed mistakes aren't simply label noise, but legitimate mistakes that leave room for improvement.

What do these lessons portend for the future of ImageNet evaluation? Top-1 will become increasingly noisy as our best models get better (though we have not yet completely saturated top-1). Our work shows that multi-label accuracy, while better at capturing "true" errors compared to top-1, suffers from a lack of a comprehensive, accurately labeled large evaluation set, which is expensive to procure and challenging to maintain.

We therefore propose ImageNet-M, a 68-example evaluation split of the multi-label evaluation set composed of "major" mistakes that several top-performing models make; we believe this subset is one on which future image classification models can achieve near perfect accuracy, and provides three clear benefits: (1) we attempt to comprehensively-label all examples for multi-label annotations to prevent the need to review novel correct predictions, (2) we endeavor to maintain and provide a way for the public to add new correct predictions; (3) the evaluation set is small enough to encourage completeness and allow the community to inspect their own errors.

## 2   Related Work

**Multi-label annotations on ImageNet.** As models continue to improve ImageNet top-1 accuracy, there has been an increased interest in evaluating ImageNet multi-label accuracy [33, 3, 31, 42, 36]. Stock et al. [33] use non-expert human studies and explanations (e.g., model criticisms) on predictions of a then-SOTA model on ImageNet, finding that machine accuracy is underestimated and advocating for multi-label evaluation. Beyer et. al. [3] introduce a set of Reassessed Labels (ReaL) for the ImageNet validation set containing multi-label annotations. The researchers first collected proposal labels from model predictions using a testbed of 19 models, and then, in order to reduce the number of predictions to review, they narrowed the set of models down to the 6 models that had the highest precision and recall above 97% on a set of 256 images reviewed by 5 experts. The top predictions from these 6 models were then reviewed by human annotators from a crowdsourcing platform. In a similar vein, Shankar et. al. [31] provide a multi-label annotation dataset for 20,000 of the 50,000 ImageNet validation images and find that roughly 20% of images have more than one valid label. To generate the annotations, the researchers first collected predictions from a testbed of  70 models for

each image and then reviewed the unique model predictions using a panel of three human experts. Additionally, human accuracy was evaluated on a subset of 1,000 images and a panel of 5 human experts reviewed human and model predictions on this subset. Tsipras et al. [36] also find that 20% of images in the validation set contain objects from multiple classes, and identify sources of ambiguous label classes. Hooker et. al [14] use human evaluations to label a subset of examples from ImageNet, finding that they contained multiple labels 40–60% of the time. More recently, Yun et al. [42] obtained pixel-level multi-label ground truths for the ImageNet *training set* using a machine annotator, and found that training a model with the dense annotations leads to small improvements in both top-1 and multi-label ImageNet accuracy.

Our paper focuses on (and adds to) the multi-label evaluation of ImageNet using expert labelers, updating the dataset to collapse classes that are overlapping or subset relationships to better capture the remaining real errors. We adopt the labeling methodology of Shankar et al. [31] and use a panel of human experts to determine the validity of novel predictions. In contrast to prior work, we re-visit all remaining model mistakes using human expert review, and categorize them by type and severity.

**ImageNet Label Error.** Multi-label annotation datasets for ImageNet (including our updated annotations) identify a set of images that have no correct ground truth label (i.e. label errors). Van Horn et. al [37] used experts from the Cornell Lab of Ornithology to estimate that at least 4% of the bird images are misclassified, and more recently Northcutt et. al [25] used MTURK workers to review algorithmically identified potential errors and found a label error rate of 5.83% in ImageNet. Lee et. al [18] sample 400 ImageNet mistakes and perform a categorization, similarly finding significant label error and label ambiguity.

**Mistake analysis.** Recent work on analyzing model mistakes on ImageNet has mainly focused on similarities or differences between mistake sets of independently trained classifiers. Mania et al. [20] found that predictions of different models on ImageNet are more similar than one would expect if the models were making mistakes independently. Geirhos et al. [9] studied a 16-class ImageNet classification task and similarly found that CNNs make remarkably consistent mistakes with one another but CNNs and humans have an error consistency that is only slightly above what can be expected from chance. In follow up work, Mania et al. [21] studied the *dominance probabilities* for pairs of ImageNet models, which capture the probability that a higher accuracy model will make a mistake on a particular image that a lower accuracy model correctly classifies. Their empirical analysis of dominance probabilities on ImageNet implies that the mistakes of higher accuracy models are typically subsets of the mistakes of lower accuracy models. However, Gontijo-Lopes et al. [10] and Andreassen et al. [1] found that training on larger and more diverse datasets as well as zero-shot evaluation can lead to models with more mistake diversity, which in turn can be used to build more accurate ensembles. Similarly, Nguyen et al. [23] found systematic differences in the errors between wide and deep ResNets on ImageNet. Ciregan et al. [4] demonstrated that powerful models on MNIST allowed for remaining error analysis in the single-label context, highlighting the ambiguity in many of those remaining errors. More recently, ImageNet-9 [40] studied the influence of background on object recognition on ImageNet, Salient ImageNet [32] annotated core and spurious features on ImageNet, and Domino [7] automatically detected systematic errors based on clustering.

In contrast to this prior work, we take a step towards fully calibrating SOTA model evaluations by exhaustively and visually reviewing every remaining model mistake using manual expert review in the *multi-label context*; we believe the remaining errors identified to be legitimately incorrect and tempered by severity and category ratings that we hope prove useful to future evaluations.

## 3 Mistake analysis method and taxonomy

To obtain an initial set of mistakes remaining on ImageNet, we used a standard ViT [6] model scaled to 3B parameters (ViT-3B) that was pre-trained on JFT-3B [34] and fine-tuned on ImageNet-1K [5], achieving a top-1 accuracy of 89.5% (details in Appendix **??**). We also later review mistakes made by the Greedy Soups model [39]. Using the `imagenet2012_multilabel` dataset [31], we measured the initial *multi-label accuracy* (MLA) of the ViT-3B model to be 96.3%. The model made a total of 676 apparent mistakes, which we sought to manually review in detail. Exhaustively examining every mistake has been made more convenient and practical due to the quality of today's top models as well as the smaller subset of 20k validation images present in the multi-label set.

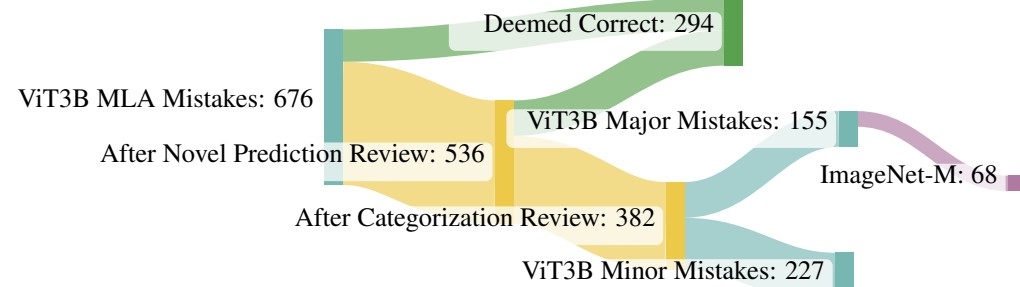

Figure 1: **Review flow**: Starting with 676 mistakes, we employed two phases of mistake review, identifying 382 re-evaluated mistakes at the end. The categorization phase identified 155 major errors made by the model. ImageNet-M was then selected from these major errors via model-based selection (Section 5.1). 294 of the original 676 mistakes were determined to be correct after review.

## 3.1 Panel Review

To exhaustively and accurately assess every remaining mistake, we formed a panel of five reviewers and followed a process similar to [31] to evaluate the predictions made by this model on the 676 mistakes — we avoided using non-expert crowd-source platforms specifically because the remaining mistakes are often difficult to assess by non-expert annotators [36, 3]. For every mistake, the panel determined: (1) Did the model make a mistake? (2) Was the original ground truth annotation correct? (3) If the model made a mistake, what is the category, type, and severity of the mistake? The `imagenet2012_multilabel` dataset contains a field for every image indicating which classes a large previous suite of models predicted that were determined to be incorrect (wrong_multi_labels). Of these 676 initial mistakes, 221 were novel: they were not reviewed in the original multi-label annotation process since none of the models evaluated made the same prediction. Each member of the panel reviewed all 221 novel mistakes.

Similar to [31], we built a review tool that allowed each panelist to see a) the predicted class, b) the predicted top softmax score, c) the set of ground-truth labels, d) the set of previously incorrect labels, and e) the image. We also employed the labeling guide produced by the authors of [31] when investigating the definition of a class, and a tool to iterate through the images of every ImageNet validation example for that class, using the validation images to help define the class boundaries (rather than the gloss or lay definition of the class). See Appendix **??** for screenshots of the review tools. In addition, we collapsed a small number of classes for which previous work has identified exhibited extreme overlap [24], such as 'missile' and 'projectile missile'. In Appendix **??** we provide these new collapsed class mappings.

We also used Google Image Search to help provide context to some assessments; in one interesting but not isolated case, a prediction of a taxi cab (with no obvious taxi cab indicators beyond yellow color) was present in the image; we determined the prediction to be correctly a taxi cab and not just a standard vehicle by identifying a landmark bridge in the background in order to localize the city, and a subsequent image search for taxis in that city yielded the images of the same taxi model and license plate design, validating the model's actually correct prediction.

Each panelist rated whether these novel mistakes had a mislabeled ground truth, or whether the prediction should be added to the set of correct, unclear, or wrong multi-labels. As a group, we reviewed any image where there was no unanimous agreement, allowing those in the minority to make their case and change minds, or highlight potential oversights. The use of a panel and discussion was important: in a non-trivial number of cases, a single panelist found unique evidence for or against a prediction that no other panelist saw that led to a different outcome. After locking in final votes, we took the majority assessment, or used 'unclear' for a tie. After this re-assessment, 140 of the novel predictions were deemed correct (or the ground truth deemed incorrect), leaving 536 remaining mistakes to assess.

## 3.2 Mistake severity and category

The remaining mistakes then comprised images that had either previously been deemed wrong by the panel in Shankar et al. [31], or deemed wrong by the current panel. We then began a review of the

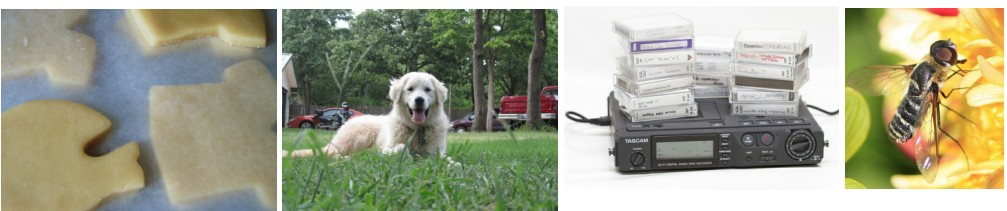

(a) **Major mistake**
Label: dough
Model: jigsaw puzzle

(b) **Minor mistake**
Label: kuvasz
Model: Great Pyrenees

(c) **New multilabel**
Label: tape player
Model: cassette

(d) **Problematic**
Label: bee
Model: fly

Figure 2: **Mistake Severity.** Examples of the two mistake severities (a-b), a correct model prediction where the model identifies a previously missing multi-label (c); and a problematic example (d) where the label (bee) is incorrect (object is a bee-fly, which is a type of fly).

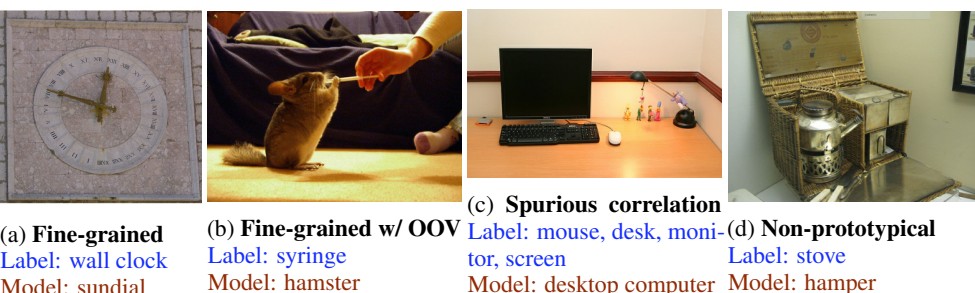

(a) **Fine-grained**
Label: wall clock
Model: sundial

(b) **Fine-grained w/ OOV**
Label: syringe
Model: hamster

(c) **Spurious correlation**
Label: mouse, desk, monitor, screen
Model: desktop computer

(d) **Non-prototypical**
Label: stove
Model: hamper

Figure 3: **Mistake Category.** Examples of the four mistake categories. In the fine-grained with OOV example, the animal is a chinchilla, which is not an ImageNet class but is visually similar to a hamster, which is an ImageNet class. In the spurious correlation example, the scene contains relevant context for desktop computer, but there is no such object in the image.

mistakes for the category and severity of mistakes. During this second phase review, we re-reviewed all images in detail, potentially overturning decisions made by the previous panel, where they missed the presence of an object that was in an example or the decision was inconsistent with the labeling guide or validation examples we relied on.[2]

**Severity:** We assessed each mistake's severity with the assumption that not all mistakes are equal: some mistakes are extremely borderline, particularly because the ImageNet class definitions are imprecise or because the image itself provides ambiguous or incomplete information. For any remaining mistake, we broke down the severity levels into **major** and **minor** mistakes. **A major mistake** is a model prediction that a human who understands the class definitions would find obviously incorrect. For example Figure 2(a) shows an example image where the prediction is jigsaw puzzle but the label is dough; although the pieces are somewhat jigsaw-puzzle-shaped, untrained humans are more likely to classify this as dough than jigsaw puzzle. **A minor mistake** is a model prediction that a human who understands the class definitions would probably find to be incorrect, but in a more subtle way than a major mistake. Some minor mistakes are so subtle that even expert-trained humans might debate their correctness.

We recognize that these severities are subject to the influences of the worldview of the panelists (and the web) and should be judged accordingly (see Section 5.3). For transparency, we provide the panel assessments of the severity for all the mistakes, and in Figure 2 we provide some examples of the severities of various mistakes made by this model for the reader's calibration. In Appendix **??** we provide many additional examples of every severity and category.

**Category:** After reviewing many mistakes, we formulated four mistake types (Figure 3).

---

[2]The labeling guide in Shankar et al. [31] was constructed after the initial panel review but prior to the human accuracy assessment, resulting in some validation labels that would be inconsistent with the labeling guide we used.

**(1) Fine-grained errors** are where the model makes a mistake between two similar types of organisms or objects, one of which is a groundtruth label. These mistakes often occur when the two confused classes are already similar (e.g. two dog breeds), or when either of them is very broad or ill-scoped (e.g. the "bake shop" class includes any baked good or bakery).

**(2) Fine-grained with out-of-vocabulary**: there is an object in the image that is not in the ImageNet class hierarchy but is similar to a predicted class that is in ImageNet. We separate out this category because it highlights the possible benefits of training models to expect new classes to appear at test time, and the importance of model uncertainty and calibration in the face of this 'open world'.

**(3) Spurious correlations**: Either (a) the predicted object is not plausibly in the image but surrounding cues may have been used, or (b) the predicted object does not match the groundtruth. In extreme cases, there is no clear indication of the predicted object; in more subtle cases, it is more clear the model is trying to predict the class of an object in the image but the predicted object would not be considered either semantically or visually similar to the groundtruth class.

**(4) Non-prototypical labels**: The predicted label is not present but the groundtruth object is a non-prototypical example of the class that bears resemblance to the predicted label. Non-prototypical mistakes are relatively rare, and capture the 'long tail' of examples for each class. These mistakes highlight the internal diversity of each class, and the difficulty of modeling the long within-class tail.

## 4 Analyzing the remaining mistakes

After review of all original 676 mistakes (comprising both novel predictions and previously reviewed mistakes), we found that a total of 298 were either correct or unclear, or determined the original groundtruth incorrect or problematic. Our evaluation of the ViT-3B model on this re-labeled dataset is shown in Table 1, with the model making a total of 378 mistakes on the dataset. In other words, approximately 44% of the initial mistakes made by this model were determined to be correct!

|  | All | | Organisms | | Objects | |
|---|---|---|---|---|---|---|
|  | MLA | MLA Re-labeled | MLA | MLA Re-labeled | MLA | MLA Re-labeled |
| ViT-3B | 96.3% | 97.9% | 96.3% | 97.8% | 96.4% | 98.0% |

Table 1: Multi-label accuracy (MLA) of ViT-3B model before and after our re-labeling.

### 4.1 Mistake category and severity

Each of the 378 remaining mistakes was assigned both a mistake category and severity (Table 2) by the expert panel.

| Model | Dataset | Categories | | | | Severities | |
|---|---|---|---|---|---|---|---|
|  |  | Fine-grained (FG) | FG w/ OOV | Spurious | Non-prototypical | Major | Minor |
| ViT-3B | ImageNet | 64.0% | 14.3% | 13.8% | 7.9% | 40.7% | 59.3% |
| ViT-3B | ImageNetV2 | 66.0% | 9.4% | 15.1% | 9.4% | 41.5% | 58.5% |
| Greedy Soups | ImageNet | 69.1% | 10.4% | 12.9% | 7.6% | 35.7% | 64.3% |

Table 2: **Mistake Category and Severity:** We classified the majority of the ViT-3B mistakes as fine-grained or a variant of fine-grained; many of the mistakes were considered "major" mistakes; these distributions held on ImageNetV2 as well as for the Greedy Soups model.

**Category:** 78.3% of errors were assessed to be fine-grained in nature (either in-vocabulary or OOV), and 13.8% categorized as a spurious correlation. To measure whether these categories are meaningful, we measured the "hierarchy distance" between the groundtruth label and the model's prediction using the WordNet hierarchy [22]. For example, a hierarchy distance of 1 means that the groundtruth and model prediction share the same parent; a distance of 2 means they share the same grandparent. We found that 80.9% of errors with a hierarchy distance of 1 were assessed as fine-grained. In contrast, 54.3% of errors with a hierarchy distance of 3 were fine-grained, matching intuition that predictions close in the hierarchy are very likely to be fine-grained errors, and predictions far in the hierarchy are more likely to be spurious correlations, fine-grained with out-of-vocabulary, or non-prototypical examples. We note that WordNet does not provide a perfect (automated) category function: "goblet" and "vase" are a hierarchy distance of 4 apart, and we encountered one model mistake for this pair that we nonetheless assessed as fine-grained.

**Analyzing class confusions:** Given that most failures were fine-grained, we tried to identify any patterns present in the class confusions made by the model, but found no consistent pattern. The inline figure shows the frequency of occurrence for confused class pairs. The distribution is long-tailed in nature—the majority of class pairs occur exactly once or twice and only a handful of class pairs occur three or more times. The long-tailed nature of the class confusions suggests that we will not be able to resolve a large fraction of model mistakes by focusing on cleaning up or adding additional data to only a small number of classes.

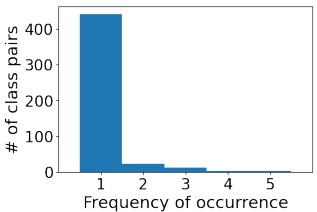

**Severity:** We determined that around 40% of errors were assessed to be "major" errors, indicating that this model still appears to make mistakes that a human familiar with the class definitions would not make, despite the fact that the model on average performs better than an expert human. We return to 'major' errors later in Section 5.1, as we believe that a subset of these errors can be a useful evaluation slice for future ImageNet benchmarking.

## 4.2 Out-of-distribution generalization

We evaluated the ViT-3B model on the ImageNetV2 multi-label subset which produced over 900 unreviewed errors. To assess what aspects of our analysis generalize to other datasets, we sampled 100 of these errors using the same panel review system employed for ImageNetV1. We discovered that 47 of the 100 ImageNetV2 predictions were either correct or had problematic labels, leaving 53 mistakes that we reviewed for category and severity. For ImageNetV1, we had previously found that 44% (296/676) of mistakes were either problematic or correct and we found no statistically significant difference between the two datasets in this regard ($\chi^2(1, N = 776) = 0.36, p = .55$). These proportions suggest that large models are frequently uncovering new correct multi-labels, suggesting that mistake analysis and label correction needs to be part of the lifecycle and maintenance of benchmark development of long-tailed errors to properly assess performance as a benchmark saturates. Table 2 compares the category and severity breakdowns between the two datasets—overall the model is making similar types and severities of mistakes on both datasets. A chi-square test of independence shows that there is no significant difference between either the mistake categorization $\chi^2(3, N = 434) = 0.97, p = .80$. or mistake severity $\chi^2(1, N = 434) = 0.01, p = .92$.

## 4.3 Generalization to new models

As models produce higher top-1 accuracy, how do the types of mistakes they make and improve upon change? We use the Greedy Soups model that obtains 90.9% top-1 accuracy on ImageNet validation [39], measuring its MLA after our initial re-labeling at 98.1%, and yielding 341 total remaining (and partially unreviewed) errors.

The Soups model corrected 209 mistakes that the ViT-3B model made, while the model made 170 mistakes where the ViT-3B model was correct, yielding an overall accuracy improvement; 28 mistake examples were common with ViT-3B but with a different prediction. In total there were 198 novel predictions made by this model that needed to be reviewed; upon review using the same panel method, we found 46.5% (92/198) were problematic (10) or actually correct (82), showing with a second model that model predictions on mistakes need to be reviewed, and that the single label expected by top-1 is often insufficient. The Soups model in the end made only 249 errors, for an MLA of 98.6%, a 0.5% absolute increase compared to unreviewed mistakes. The categorization and severity of these errors is shown as the last line in Table 2. Again, a chi-square test of independence shows that there is no significant difference between either the mistake categorization ($\chi^2(3, N = 629) = 2.41$, $p = .49$) or mistake severity ($\chi^2(1, N = 629) = 1.61, p = .20$).

## 4.4 Comparison to humans

We compare the performance of the ViT-3B and Greedy Soups models to the best human labeler from Shankar et al. [31][3] by evaluating on the subset of 1,000 ImageNet val images used to compute human accuracy in the prior work. To fairly compare to the models, we re-compute the human accuracy

---

[3]Best human is chosen based on MLA on all classes (as opposed to object or organism subset) and corresponds to Human E in the original work.

scores using the original human predictions and our updated label set. Overall, the re-labeling did not significantly change human accuracy; the best human labeler achieved 97.3% MLA on the original multi-labels and 97.4% MLA on the updated multi-labels.

Table 3 compares the re-labeled MLA of the ViT-3B and Greedy Soup model to the best human for all ImageNet classes as well as the subset of ImageNet classes corresponding to objects and organisms. Similar to Shankar et al. [31], we also find that the performance of the models is more uniform across the object and organism classes, but humans do substantially better on the object classes than the organism classes. However, unlike prior work, current models outperform the best human when evaluated on all ImageNet classes (though humans still achieve slightly better performance on the object classes).

| | All Classes | Organisms | Objects |
|---|---|---|---|
| ViT-3B | 97.8% | 97.4% | 98.1% |
| Greedy Soup | 98.6% | 98.4% | 98.8% |
| Best Human [31] | 97.4% | 95.4% | 99.0% |

Table 3: **Multi-label accuracy compared to humans**. Both the ViT-3B and Greedy Soup model achieve better MLA on all ImageNet classes than the best human labeler from Shankar et al. [31]. However, on the object classes, where the class boundaries are substantially less ambiguous for humans, the best human labeler still outperforms the models.

### 4.5 Analyzing the Training Data

Finally, we investigate how much we can understand model validation set errors through the lens of the training data. To do so, we inspect the K = 10 nearest neighbor training images using JFT-pretrained (before ImageNet fine tuning) embeddings for the ViT-3B model. Doing this, we rediscover (originally documented in Sun et al. [34] and Kolesnikov et al. [16]) that 797 (1.59%) ImageNet validation images exist in the training set as **exact duplicates**. Interestingly, we are the first to notice that **every single leaked image has a different label in the training set than the validation set**. In Appendix **??** we show removing these images has relatively little impact on the model's performance, and detail a more pernicious leakage pattern of "near duplicates" in Appendix **??** that is hard to fully quantify. Finally, we show how the training data sometimes explain spurious correlations in Appendix **??**.

## 5 Recommendations and Discussion

In this section we provide recommendations for future evaluation, starting with ImageNet-M: our curated multi-label evaluation set of "major mistakes" that we suggest should be reported on in addition to metrics such as top-1 and multi-label accuracy.

### 5.1 ImageNet-M: A "major mistakes" evaluation split

Studies over the last several years seeking to understand whether ImageNet remains an informative benchmark have typically concluded that aspects of ImageNet remain useful but top-1 accuracy less so. These works encourage alternative related metrics for ImageNet such as multi-label accuracy [3, 31] or object vs. organism breakdowns [31]. In many of these cases, stronger, more generalizable models often continue to incrementally but inevitably improve on these metrics.

With an emphasis on understanding which long-tail errors are unambiguously errors, we suggest that a benchmark focusing on the most egregious long-tail errors could provide a useful additional signal about whether the improvements are meaningful. In particular, we desire a long-tailed benchmark where we believe that 100% accuracy is achievable. Because the minor mistakes are more subject to interpretation and discussion than the major mistakes, we believe a benchmark focused on the latter will help the community judge what is meaningful improvement on ImageNet.

To that end, we leverage our expert-reviewed analysis to produce a small slice of the ImageNet multi-label set where (1) today's *best* top-1 models are still more wrong than right, and (2) the mistakes are largely unambiguous to a human given a reasonable understanding of the ImageNet label set. We call this evaluation slice **ImageNet-M**ajor.

**ImageNet-M example selection method.** The ViT-3B model made 155 "major" mistakes, for which we analyzed whether each example was labeled correctly for three additional models: (1) the Greedy Soups model, (2) a model pre-trained on Instagram data but fine-tuned on ImageNet that achieves 85.4% top-1 [19], and (3) A zero-shot evaluation [28, 15, 27] using a CoCa [41] model pretrained on JFT and noisy image-text data. In order to maximize prediction diversity, we purposefully selected models with varying pre-training data and training methodologies, including a zero-shot model that does not see ImageNet image-label associations directly [10].

From this suite of four models, we assembled a subset where three or more of the models make a major mistake, yielding 68 such major mistakes. This process is similar in spirit to ImageNet-A [13], except we use four high-performing but diverse models, and we restrict the set to ImageNet images and the corresponding model prediction that were rated as "major" errors. We analyzed the predictions of all models on these examples (including any novel predictions made by these additional models) and verified that none of them were correct new multi-labels, and that any model's mistakes were major mistakes. In addition, we attempted to comprehensively manually label additional labels that no model has yet predicted but would be correct, in an attempt to reduce the likelihood that future models are penalized for making novel but unreviewed correct predictions. The failure categories of these 68 examples are broken down into 36 fine-grained failures, 15 spurious correlation failures, 12 out-of-vocabulary failures, and 5 non-prototypical failures.

We design the ImageNet-M 68-example subset as an additional split of the validation set that we believe has the following properties: (a) many top performing models trained in different ways make mistakes on this set; (b) the mistakes are all major mistakes as determined by expert-reviewers; (c) the example set is small enough to permit manual inspection by model evaluators; (d) strives to be comprehensively labeled with respect to the ImageNet label set; (e) in theory provides a subset that *future models could achieve perfect accuracy on* without worrying about underspecified class definitions. We anticipate stronger models may yet make correct unreviewed predictions on this slice, so we will endeavor to update the set of multi-labels as they are reported to us.

**Evaluation.** By construction, our ViT-3B model achieves 0% accuracy on ImageNet-M; the Instagram-pretrained model gets 9 of the 68 correct, while the Greedy Soups model gets 19 correct. The zero-shot model gets the best performance with 24 correct, even though the zero-shot model overall achieves lower multi-label accuracy (94.2%) than any of the other models. Because we use these four models to help choose the mistakes, these specific models comparatively will perform poorly on this benchmark. Like with ImageNet-A [13], a dataset chosen using models may bias selection in such a way that future models may easily get very high accuracy on this subset, though we try to mitigate this effect by using multiple differently-trained models in our selection criteria; we discuss this selection criteria in more detail in Appendix **??**.

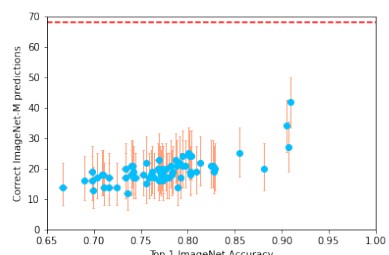

Trend of models on top-1 ImageNet vs. ImageNet-M, using Clopper-Pearson intervals. Red dashed line indicates total number of images in ImageNet-M as an upper-bound.

How do models that were not used to select this dataset perform? We evaluate the suite of 70 models from Shankar et al. [31] on this dataset, in addition to four recent top models not directly used to help filter the ImageNet-M set: a ViT-G/14 model [43] (90.5% top-1), a BASIC model [27] fine-tuned on ImageNet (90.7% top-1), an ALIGN model [15] fine-tuned on ImageNet (88.1% top-1), and a CoCa model [41] fine-tuned on ImageNet (91.0% top-1). The plot shown here shows that most models as far back as AlexNet through ResNets get between 10-25 examples correct, but recent high accuracy models such as ViT-G/14, BASIC-FT, and CoCa-FT are starting to solve more of these 'major' mistakes: CoCa-FT gets 42 of the 68 examples correct. We reviewed the mistakes made by these four new models, which yielded a total of 5 novel predictions; 4 of them were verified to be wrong (and major), and 1 additional new valid prediction, for which we updated the label set accordingly. Future models could predict classes on these examples that are "minor" mistakes, since the definition of severity is linked to the (prediction, example) pair; the dataset slice could be augmented with an attribute to provide more fine-grained signals.

## 5.2 Evaluation recommendations.

One might view ImageNet-M as another quantitative hill-climbing target, but while benchmarks like this invite quantitative comparison, we caution against using ImageNet-M as a purely quantitative benchmark for a few reasons.

First, the evaluation set is relatively small; error bars will be wide enough that it will be hard to claim one model is quantitatively better than another. Second, ImageNet-M is predominantly qualitative in nature; the relatively small evaluation set by design permits qualitative analyses of the results that a much larger set would effectively prohibit. Hypothetically, assume two models $A$ and $B$ where $A$ gets 40 of the 68 correct and $B$ gets 55 of the 68 correct, but where all of the mistakes model $A$ makes are minor/borderline, and all of the mistakes $B$ makes are still major mistakes. Qualitatively, model $A$ might be the better model for some use cases because it makes no major mistakes. To even understand whether a mistake is major or minor today requires human analysis, which is why ImageNet-M is reasonably small.

Our overall recommendations would be to look at model mistakes on ImageNet-M *qualitatively*: (1) Check that the mistakes are actually mistakes, and if not, update ImageNet-M labels. (2) Check that the mistakes are major; for those that are, identify what error category the mistake belongs in, and apply some appropriate remediation. A model that does not improve upon accuracy, but makes fewer (or no) "major" mistakes, could still be very noteworthy. (3) As one iterates on a model, evaluating on ImageNet-M to understand how their changes changed the distributions of mistakes a model is making could provide additional insight into whether progress is being made on major mistakes.

Lastly, although we did not find statistical evidence that stronger models solve major errors first, we hope progress on image classification can be evaluated by whether improvements help reduce egregious mistakes before reducing nebulous ones. Overall, we encourage reporting on several slices, including ImageNet-M, to measure the strengths and weaknesses of various models.

## 5.3 Limitations of Analysis

Much of our analysis on mistake categorization, severity, and data cleaning depends greatly on qualitative factors determined by the authors and experiment design, which we briefly discuss here.

**Limitations due to mistake subset.** We only reviewed multi-label annotation examples where the ViT-3B model and Greedy Soups model we chose were incorrect; while the multi-label dataset itself has undergone review of mistakes from a suite of many models, we never review any validation image whose groundtruth might be wrong, if all models evaluated also make the incorrect groundtruth prediction. We do partially review some of the mistakes made by a few other models, and build upon a dataset where mistakes made by many other models have already been reviewed, but most of the work here assesses these two specific model's predictions.

**Definition of a mistake.** We re-iterate that we used qualitative judgments to decide whether a prediction was a mistake, and if so, its severity and categorization. Our qualitative judgments are therefore based on a biased worldview [35, 8, 2, 26] comprising the five panelists; moreover, we are not world experts on dog or animal species, though we believe our assessments are at least as good or better than the original labeling process used for the validation set, given the research effort we made on each mistake. As a mitigation against imbuing too biased a worldview on class definitions, we relied heavily on the validation data to define the boundaries of the class, even when those examples did not match up with our personal definitions of the class. Moreover, updating (and evaluating) multi-labels potentially allows for different world-views to be expressed.

## Acknowledgements

We thank the reviewers for their helpful comments and discussion that improved this paper. We thank Lucas Beyer, Zhifeng Chen, Aleksandra Faust, Wei Han, Alexander Kolesnikov, Simon Kornblith, Jiquan Ngiam, Abhijit Ogale, and Jiahui Yu for their comments, suggestions, and infrastructure support. Additionally, we thank the larger Google Research organization for their support.

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

## Author Contributions

All authors contributed to reviewing and categorizing ImageNet mistakes, participated in panel discussions and conducted analysis, and helped write the paper.

VV initiated the project, built review and analysis tools, and helped design and implement experiments and various analyses.

BC performed the K-nearest neighbor error and duplicate analysis (and built tools to do so), helped write the paper, and contributed to the discussions on analysis and project direction.

RGL performed comparisons to human-evaluated labels, helped write the paper, and contributed to discussions on analysis and project direction.

SFK helped with writing and figure generation, and contributed to discussions on analysis and project direction.

RR helped assemble team, provided expertise on previous work related to labeling `imagenet2012_multilabel` dataset, contributed to writing, advised on high level research direction and experiments to conduct.

