# Supplementary Materials

## A    ViT-3B model details

The ViT model we use in this work is based on a standard Vision Transformer [7] model scaled to nearly 3 billion parameters, using a patch size of 14, 16 heads, 64 blocks, an MLP dimension of 8192 and a hidden dimension of 2048. The model is defined and trained in Lingvo [32]; we additionally employ GSPMD [41] for training. The model is pre-trained on JFT-3B [35] using training settings that optimize for performance on JFT-3B rather than for fine-tuning on ImageNet; notably, we do not use the training recipe that helps few-shot transfer performance [44]. For fine-tuning on ImageNet, we use the AdamW optimizer (beta1=0.9, beta2=0.999, epsilon=1e-8, weight_decay=0.3) with a cosine learning rate schedule (max learning rate of 3e-2, warmup of 2k steps, final rate of 3e-4), a training batch size of 512, and fine-tune for a total of 10 epochs.

## B    Review tools

We include screenshots of the reviewing tools we built to analyze model mistakes. Figure 3 shows the UI for reviewing model predictions and Figure 4 shows the UI that displays the labeling guide and slide bar to browse images for a particular class.

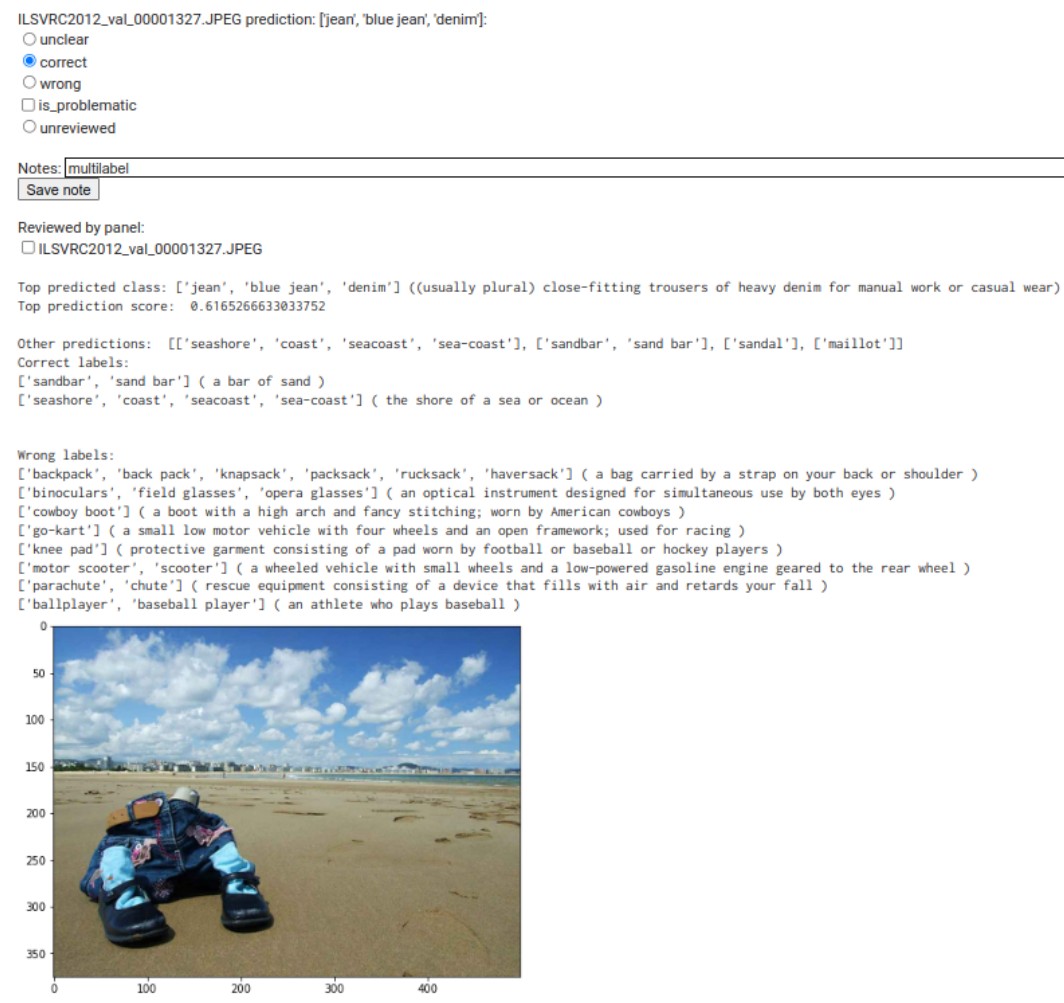

Figure 3: A screenshot of the UI we built to review model predictions. For each image, we determined whether the prediction was correct, wrong, or unclear. We also flagged images as problematic if the ground truth label for the image was incorrect.

## bakery, bakeshop, bakehouse

- The image does not have to show the building.
- Many **bakery** images have many baked goods, e.g., an entire tray of cupcakes.
- Cakes appear individually more often in this class. There is also no other class for cake, so we count cakes as **bakery**.
- More generally, if the image shows an individual baked item and there is no other suitable baked good class for it, the image counts as **bakery**.

▶ Display images

**show_n:** 50

**dataset:** INetVal

**Show code**

a workplace where baked goods (breads and cakes and pastries) are produced or sold - Key: 2136211865742104015-n02776631-ILSVRC2012_val_00010307.JPEG, is_problematic: NO

Figure 4: A screenshot of the class search tool we built that displays the labeling guide and a slider bar that allows users to browse validation images for a particular class.

# C Collapsed mappings

We provide our collapsed class mappings that we employed unilaterally, based on determining that the classes exhibited significant overlap based on the validation set images, or there was a strict superset relationship between two or more classes. For example, a prediction of 'eskimo dog' for a 'siberian husky' label would be considered correct, whereas a prediction of 'siberian husky' for an 'eskimo dog' label might not.

All siberian huskies and malamutes are also eskimo dogs. 250: 248, 249: 248

Sunglass and sunglasses are the same class (bidirectional). 836: 837, 837: 836

Indian and African elephants are also tuskers. 385: 101, 386: 101

A coffee mug is also a cup. 504: 968

Maillot and maillot, tanksuit are the same class (bidirectional). 638: 639, 639: 638

Missile and projectile missile are the same class (bidirectional). 657: 744, 744: 657

Notebook computer and laptop are the same class (bidirectional). 620: 681, 681: 620

Monitor and screen are the same class (bidirectional). 664: 782, 782: 664

A cassette player is also a tape player. 482: 848

Weasel, polecats, black-footed ferrets, and minks are all the same class. 356: [357, 358, 359], 357: [356, 358, 359], 358: [356, 357, 359], 359: [356, 357, 358],

All bathtubs are tubs, but not all tubs are bathtubs. 435: 876

 # D  Mistake Examples by Severity

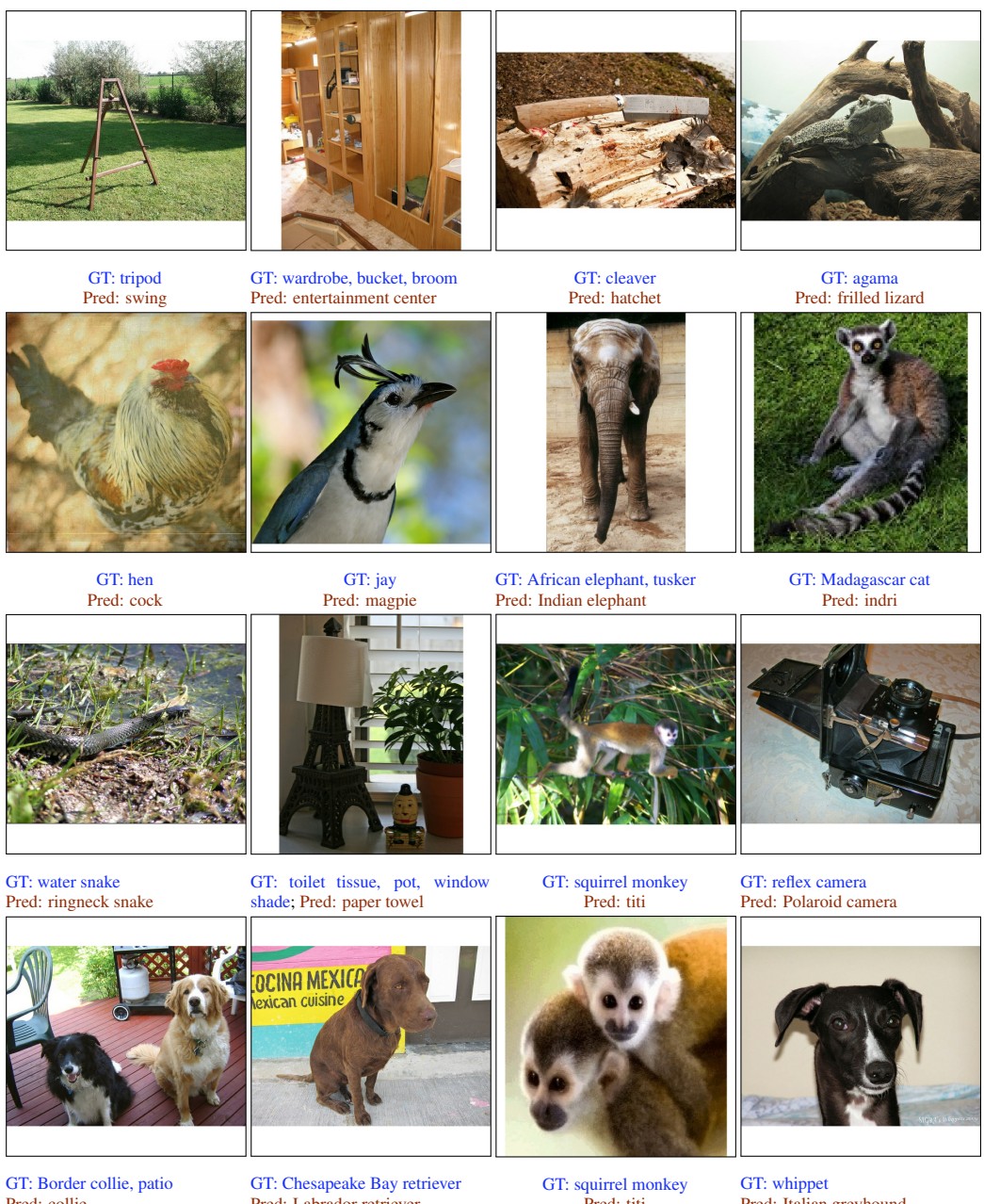

GT: tripod
Pred: swing

GT: wardrobe, bucket, broom
Pred: entertainment center

GT: cleaver
Pred: hatchet

GT: agama
Pred: frilled lizard

GT: hen
Pred: cock

GT: jay
Pred: magpie

GT: African elephant, tusker
Pred: Indian elephant

GT: Madagascar cat
Pred: indri

GT: water snake
Pred: ringneck snake

GT: toilet tissue, pot, window shade; Pred: paper towel

GT: squirrel monkey
Pred: titi

GT: reflex camera
Pred: Polaroid camera

GT: Border collie, patio
Pred: collie

GT: Chesapeake Bay retriever
Pred: Labrador retriever

GT: squirrel monkey
Pred: titi

GT: whippet
Pred: Italian greyhound

Figure 5: **Major mistakes.** Additional examples of major mistakes. Of the correct multi-labels, the original ImageNet label is listed first.

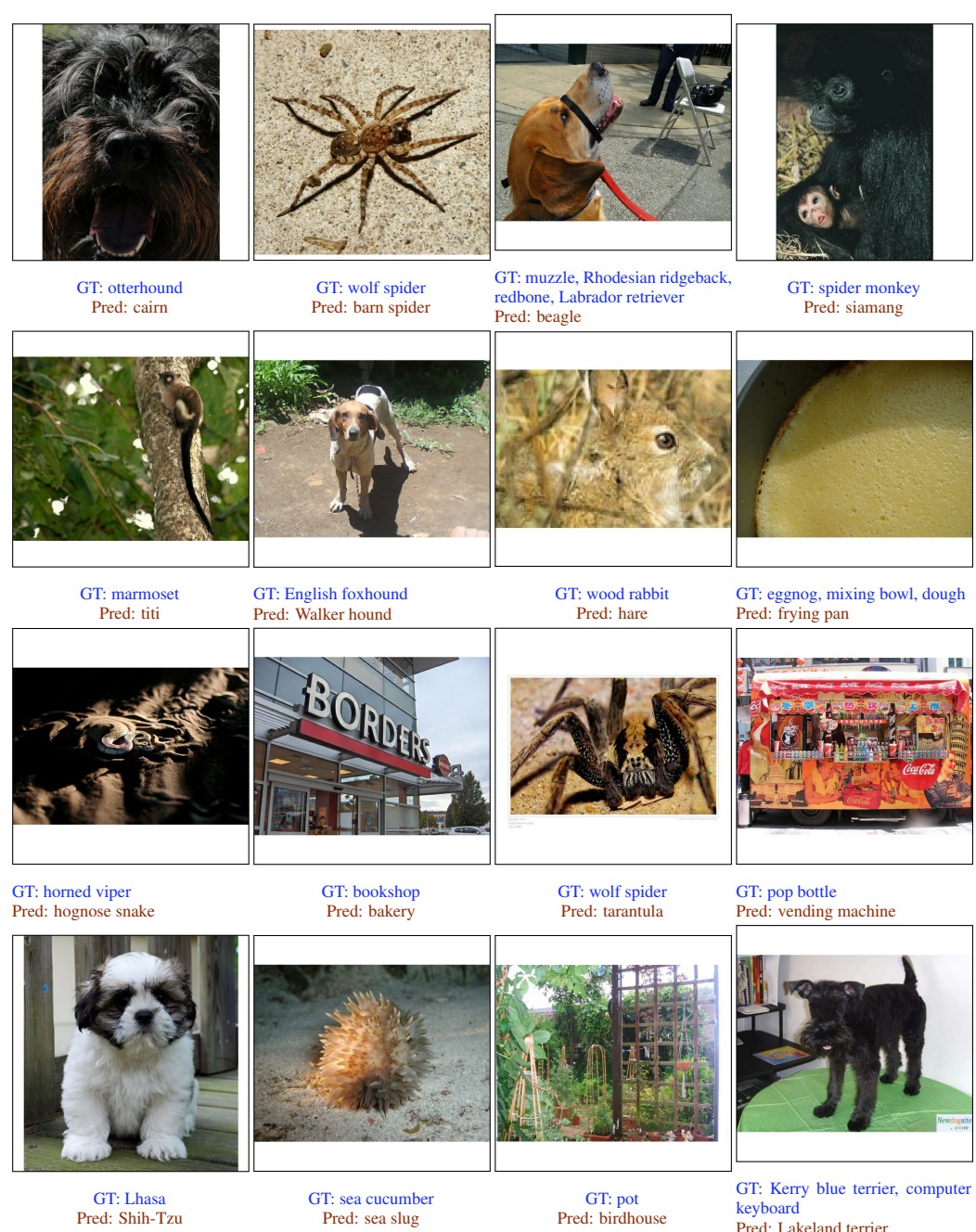

GT: otterhound
Pred: cairn

GT: wolf spider
Pred: barn spider

GT: muzzle, Rhodesian ridgeback, redbone, Labrador retriever
Pred: beagle

GT: spider monkey
Pred: siamang

GT: marmoset
Pred: titi

GT: English foxhound
Pred: Walker hound

GT: wood rabbit
Pred: hare

GT: eggnog, mixing bowl, dough
Pred: frying pan

GT: horned viper
Pred: hognose snake

GT: bookshop
Pred: bakery

GT: wolf spider
Pred: tarantula

GT: pop bottle
Pred: vending machine

GT: Lhasa
Pred: Shih-Tzu

GT: sea cucumber
Pred: sea slug

GT: pot
Pred: birdhouse

GT: Kerry blue terrier, computer keyboard
Pred: Lakeland terrier

Figure 6: **Minor mistakes.** Additional examples of minor mistakes. Of the correct multi-labels, the original ImageNet label is listed first.

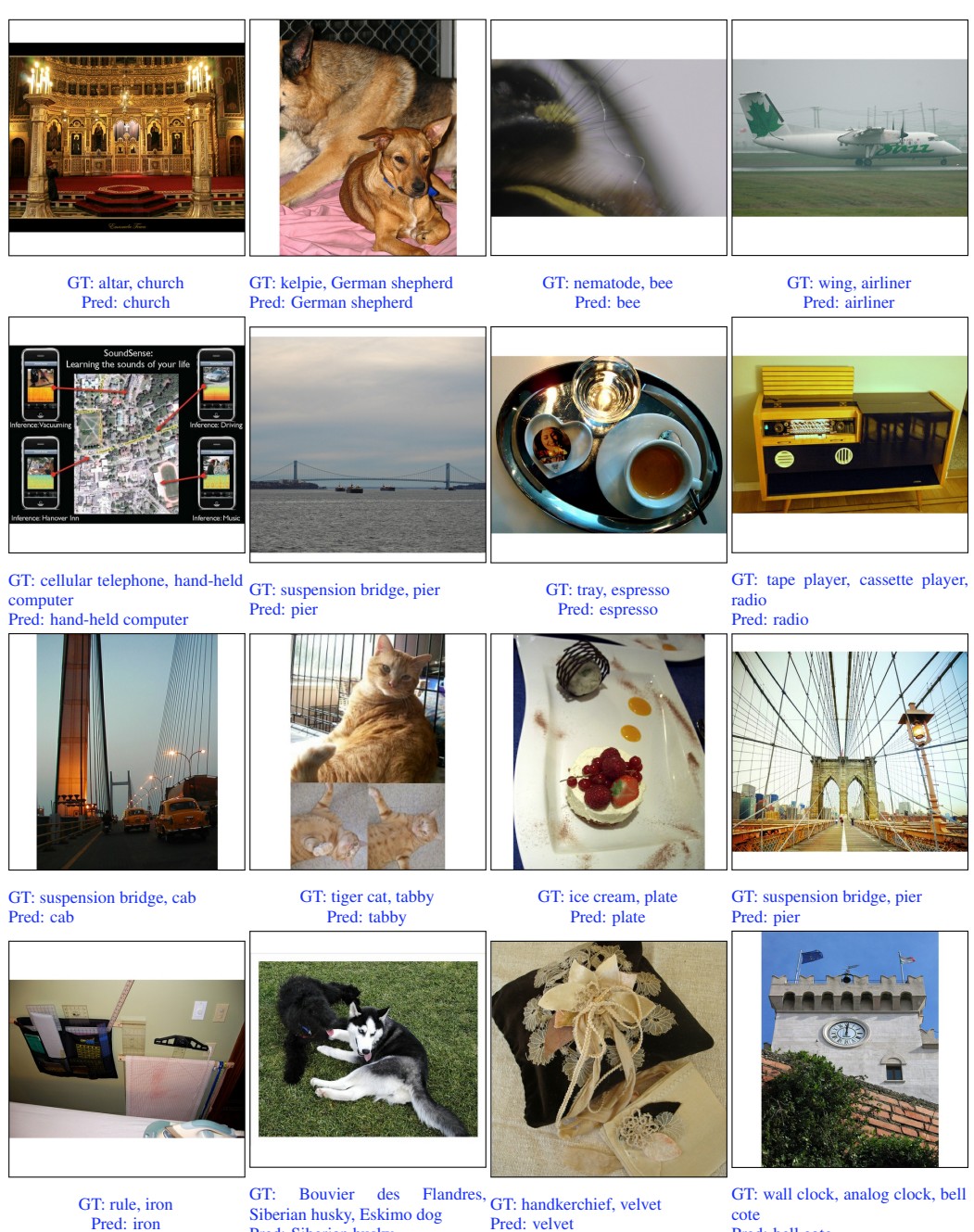

GT: altar, church
Pred: church

GT: kelpie, German shepherd
Pred: German shepherd

GT: nematode, bee
Pred: bee

GT: wing, airliner
Pred: airliner

GT: cellular telephone, hand-held computer
Pred: hand-held computer

GT: suspension bridge, pier
Pred: pier

GT: tray, espresso
Pred: espresso

GT: tape player, cassette player, radio
Pred: radio

GT: suspension bridge, cab
Pred: cab

GT: tiger cat, tabby
Pred: tabby

GT: ice cream, plate
Pred: plate

GT: suspension bridge, pier
Pred: pier

GT: rule, iron
Pred: iron

GT: Bouvier des Flandres, Siberian husky, Eskimo dog
Pred: Siberian husky

GT: handkerchief, velvet
Pred: velvet

GT: wall clock, analog clock, bell cote
Pred: bell cote

Figure 7: **Correct "mistakes".** Additional examples where the model makes a correct prediction that we add to the original multi-label annotations. Of the original multi-labels shown, the original ImageNet label is listed first. Often the model correctly identifies a different object in the image, and in some cases a single object has ambiguous class membership and could plausibly belong to either the ground truth or the predicted class.

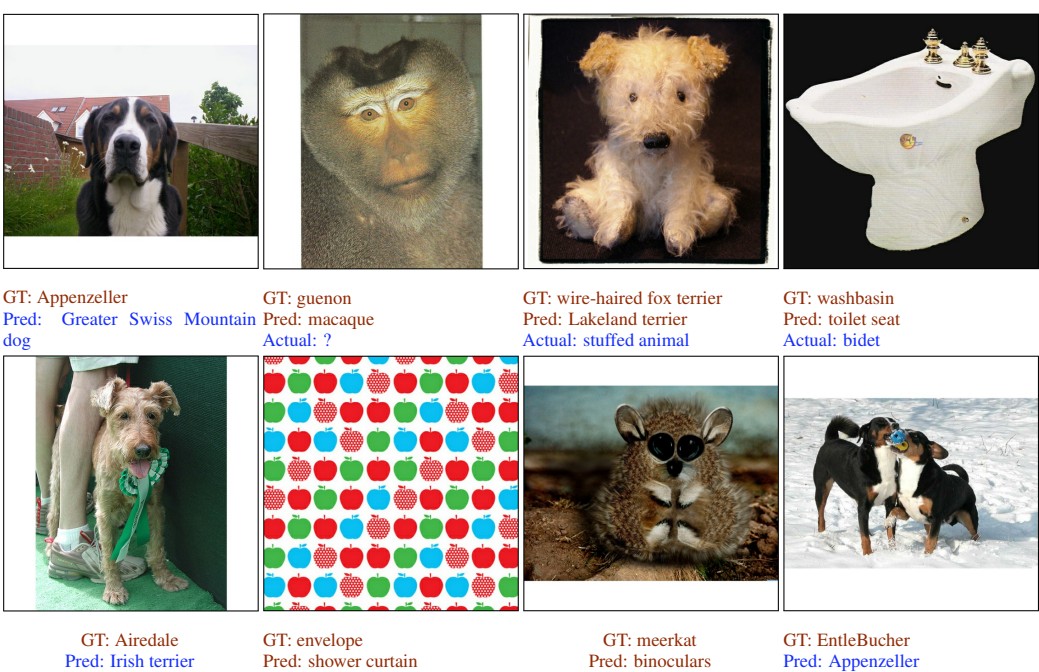

GT: Appenzeller
Pred: Greater Swiss Mountain dog

GT: guenon
Pred: macaque
Actual: ?

GT: wire-haired fox terrier
Pred: Lakeland terrier
Actual: stuffed animal

GT: washbasin
Pred: toilet seat
Actual: bidet

GT: Airedale
Pred: Irish terrier

GT: envelope
Pred: shower curtain

GT: meerkat
Pred: binoculars

GT: EntleBucher
Pred: Appenzeller

Figure 8: **Problematic "mistakes".** Examples where our panel determined that the image or its original label was problematic (and therefore should not be in the validation set). Most problematic examples are problematic because the original ImageNet label was deemed incorrect because the prediction by the model was indeed correct.

## E Mistake Examples by Category

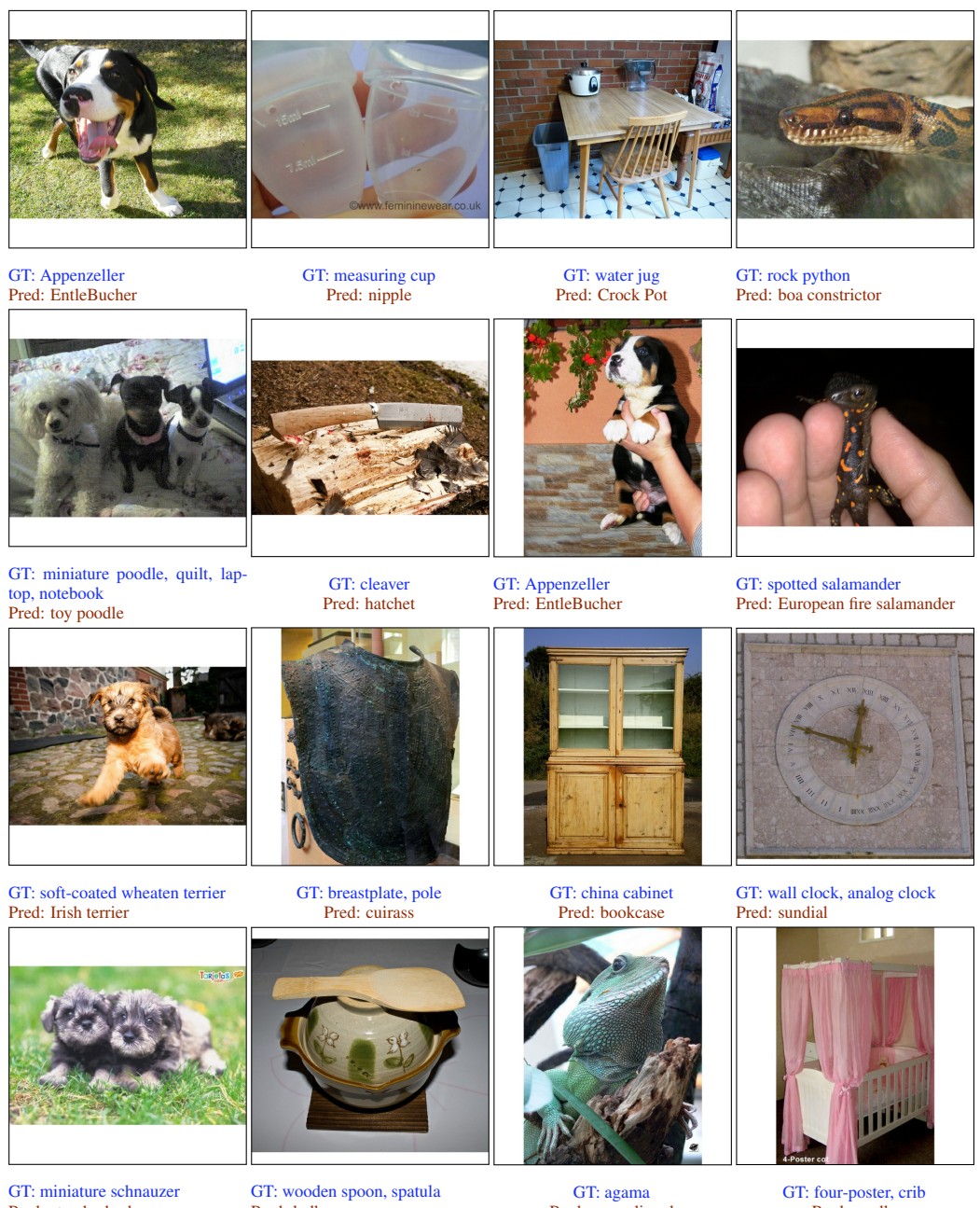

GT: Appenzeller
Pred: EntleBucher

GT: measuring cup
Pred: nipple

GT: water jug
Pred: Crock Pot

GT: rock python
Pred: boa constrictor

GT: miniature poodle, quilt, laptop, notebook
Pred: toy poodle

GT: cleaver
Pred: hatchet

GT: Appenzeller
Pred: EntleBucher

GT: spotted salamander
Pred: European fire salamander

GT: soft-coated wheaten terrier
Pred: Irish terrier

GT: breastplate, pole
Pred: cuirass

GT: china cabinet
Pred: bookcase

GT: wall clock, analog clock
Pred: sundial

GT: miniature schnauzer
Pred: standard schnauzer

GT: wooden spoon, spatula
Pred: ladle

GT: agama
Pred: green lizard

GT: four-poster, crib
Pred: cradle

Figure 9: **Fine-grained mistakes.** Additional examples of fine-grained mistakes. Of the correct multi-labels, the original ImageNet label is listed first.

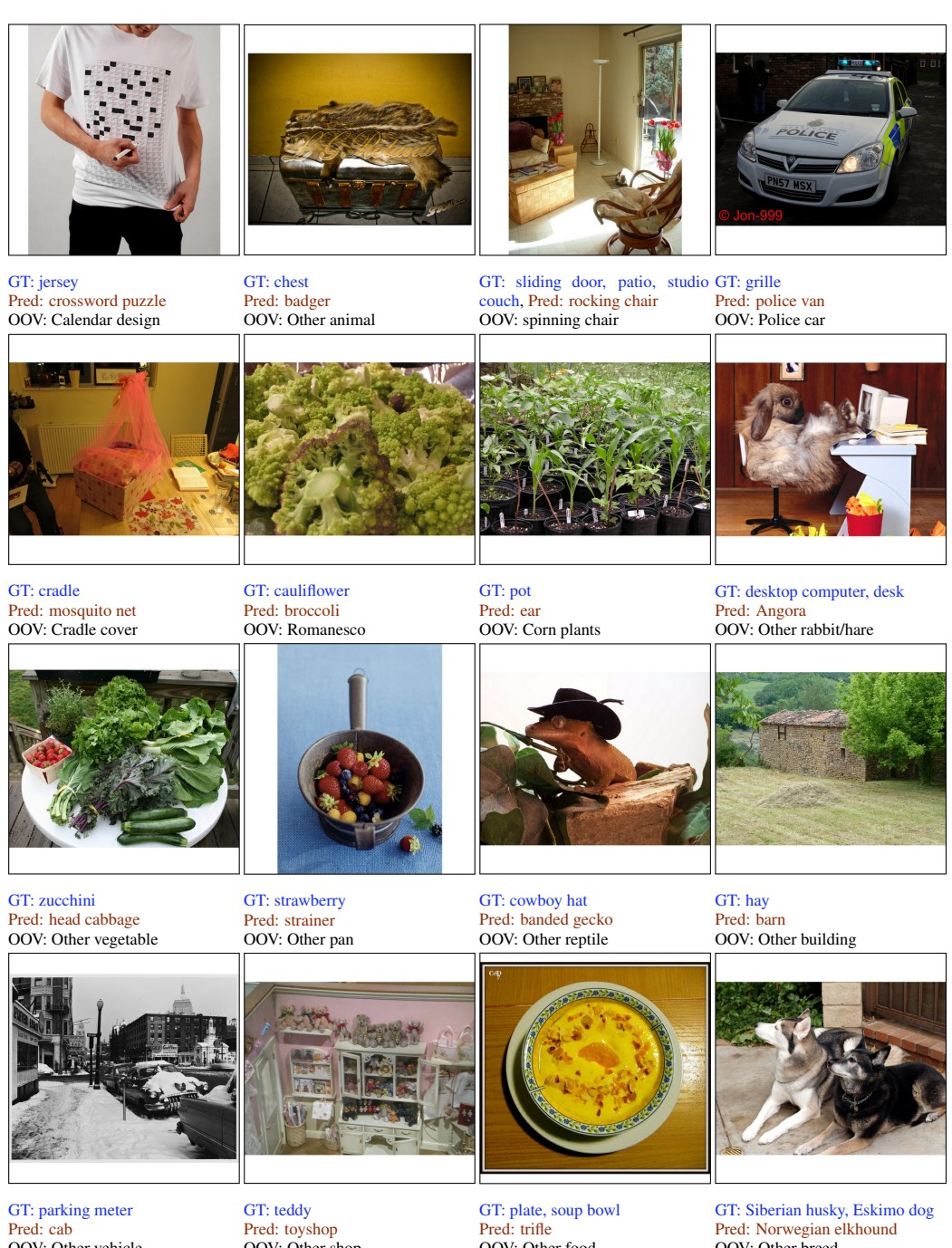

GT: jersey
Pred: crossword puzzle
OOV: Calendar design

GT: chest
Pred: badger
OOV: Other animal

GT: sliding door, patio, studio couch, Pred: rocking chair
OOV: spinning chair

GT: grille
Pred: police van
OOV: Police car

GT: cradle
Pred: mosquito net
OOV: Cradle cover

GT: cauliflower
Pred: broccoli
OOV: Romanesco

GT: pot
Pred: ear
OOV: Corn plants

GT: desktop computer, desk
Pred: Angora
OOV: Other rabbit/hare

GT: zucchini
Pred: head cabbage
OOV: Other vegetable

GT: strawberry
Pred: strainer
OOV: Other pan

GT: cowboy hat
Pred: banded gecko
OOV: Other reptile

GT: hay
Pred: barn
OOV: Other building

GT: parking meter
Pred: cab
OOV: Other vehicle

GT: teddy
Pred: toyshop
OOV: Other shop

GT: plate, soup bowl
Pred: trifle
OOV: Other food

GT: Siberian husky, Eskimo dog
Pred: Norwegian elkhound
OOV: Other breed

Figure 10: **Fine-grained with OOV mistakes.** Additional examples of fine-grained with out-of-vocabulary mistakes. Of the correct multi-labels, the original ImageNet label is listed first.

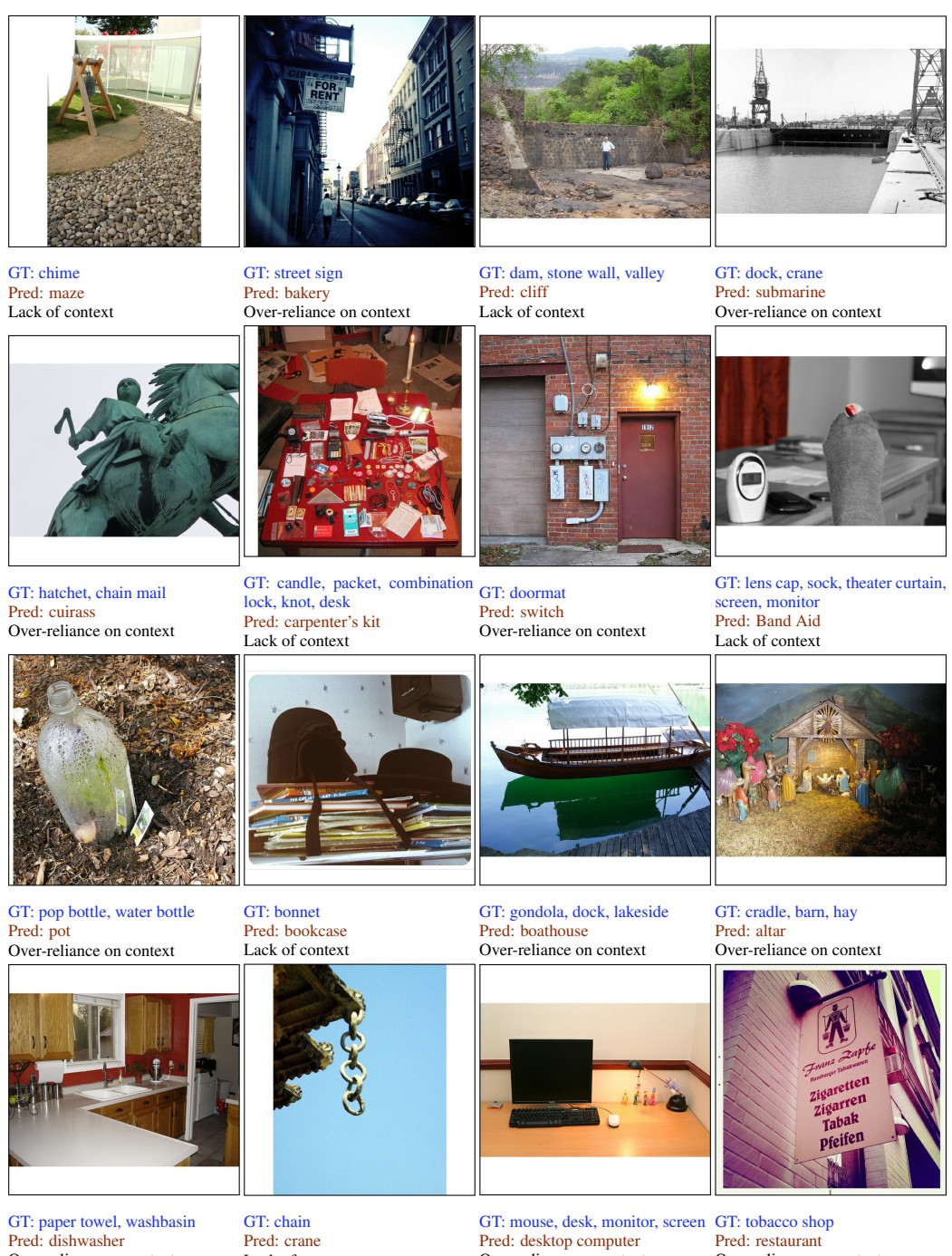

Figure 11: **Spurious correlation examples**. Of the correct multi-labels, the original ImageNet label is listed first. Over-reliance on context indicates that surrounding cues in the image correlate with the predicted class, although the predicted class is not present. Lack of context indicates that the model has failed to understand relevant context in the image, and predicts a class that is inconsistent with a holistic understanding of the image.

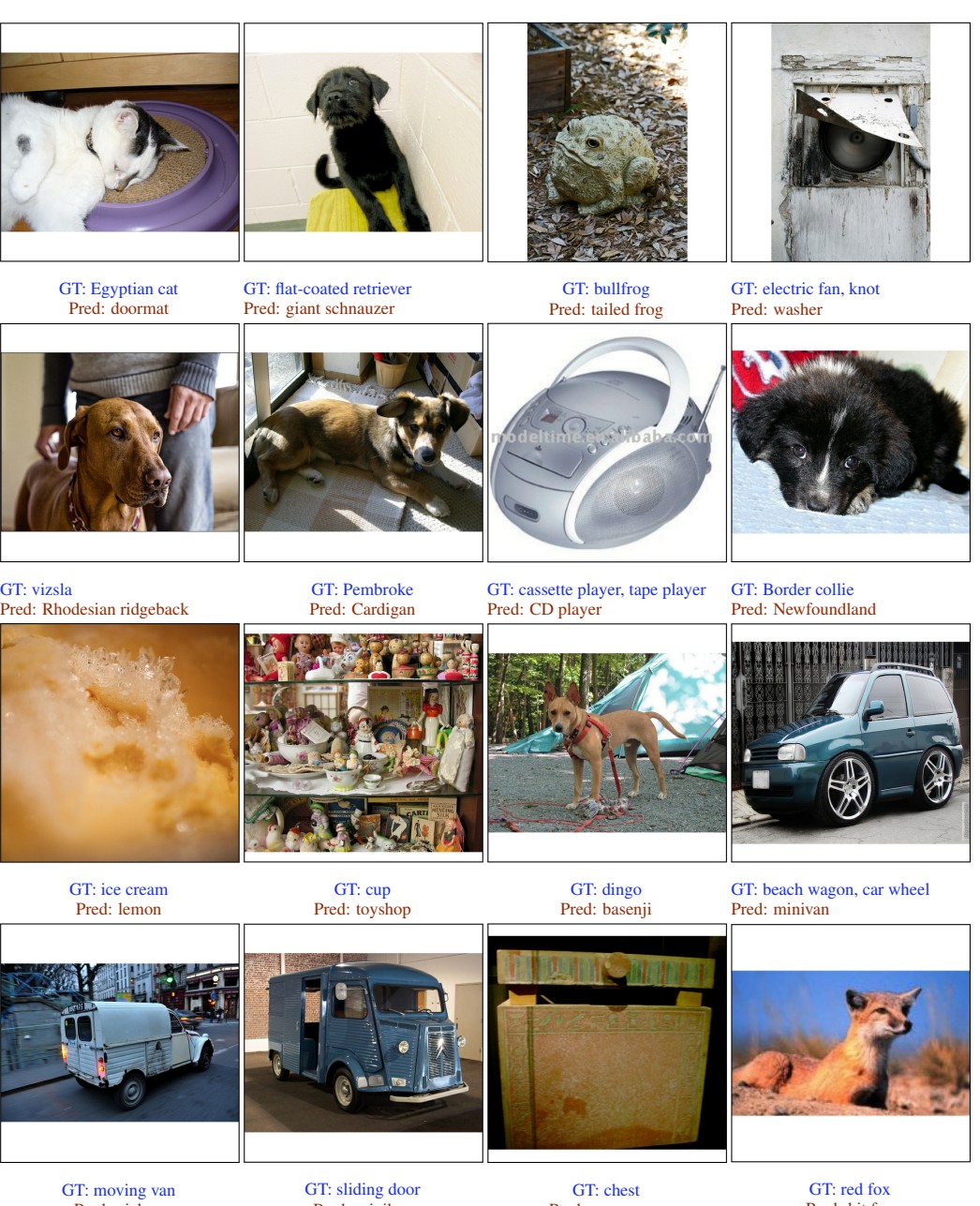

GT: Egyptian cat
Pred: doormat

GT: flat-coated retriever
Pred: giant schnauzer

GT: bullfrog
Pred: tailed frog

GT: electric fan, knot
Pred: washer

GT: vizsla
Pred: Rhodesian ridgeback

GT: Pembroke
Pred: Cardigan

GT: cassette player, tape player
Pred: CD player

GT: Border collie
Pred: Newfoundland

GT: ice cream
Pred: lemon

GT: cup
Pred: toyshop

GT: dingo
Pred: basenji

GT: beach wagon, car wheel
Pred: minivan

GT: moving van
Pred: pickup

GT: sliding door
Pred: minibus

GT: chest
Pred: prayer rug

GT: red fox
Pred: kit fox

Figure 12: **Non-prototypical mistakes.** Additional examples of non-prototypical mistakes. Of the correct multi-labels, the original ImageNet label is listed first. Non-prototypical examples are typically unusual border cases of the groundtruth class, such as puppies of a dog breed, or unusual/unique versions of the class.

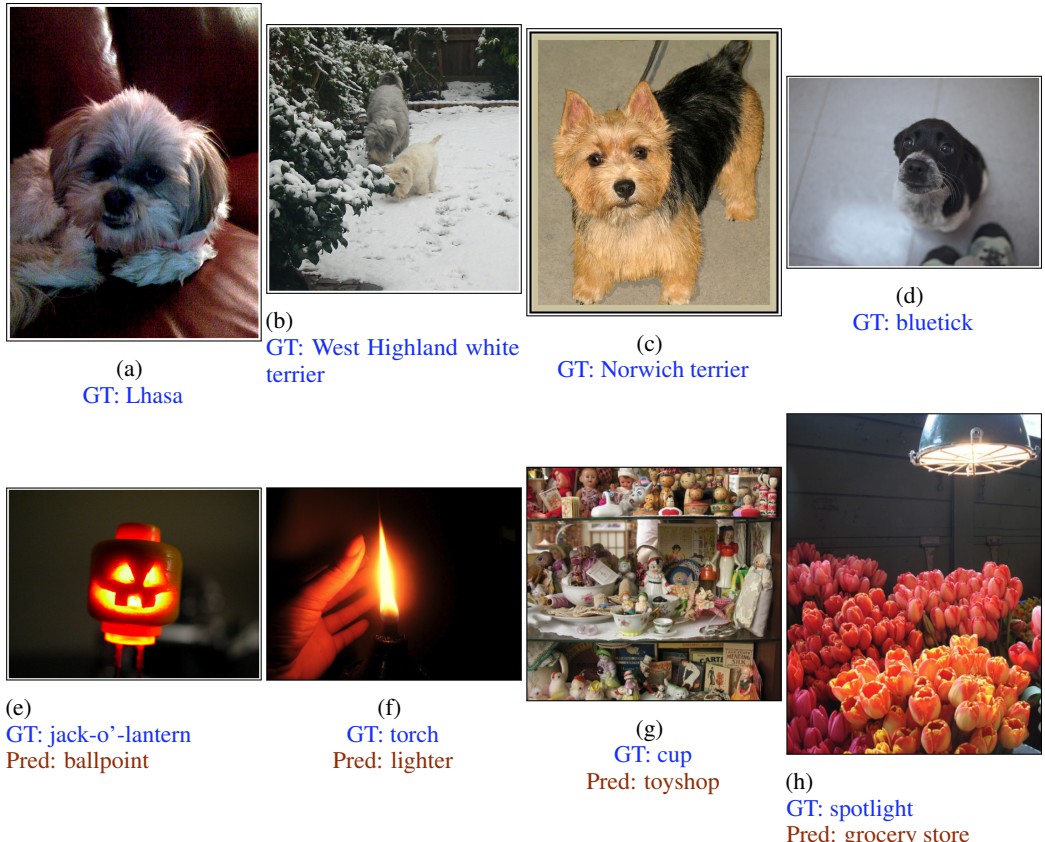

Figure 13: **Difficult images for humans and models.** Top row: the only 4 images that all humans [31] classify incorrectly (our model classifies these correctly). Bottom row: images that should-be-easy (all humans get correct), but the model gets incorrect.

## F  Confused classes

We provide a list of confused classes and their corresponding frequency of occurrence for the ViT-3B model on ImageNet validation set. TODO(ALL): Add link to public json file.

## G  Analyzing the Training Data

In this section, we try to understand the model's remaining mistakes by investigating the training data. We do this through the lens of looking at the nearest training examples to the ViT-3B models remaining mistakes. To do this, we generate embeddings for every training and validation example using the pretrained ViT-3B models JFT checkpoint (before fine-tuning on Imagenet), and for every error, we query the K=10 nearest neighbors using an exact nearest neighbor lookup.

### G.1  Validation Set Leakage.

One of the most interesting findings using nearest neighbors was rediscovering that 797 ImageNet validation images (1.594%) **exist in the training set as exact duplicates** (pixelwise L2 distance of 0), 34 of them more than once for a total of 831 duplicate training images. While this was previously documented in [34] and [16], we are the first to notice that every single leaked sample has a different label in the train set than in the validation set, indicating the ImageNet authors did de-duplicate within a class, but not across classes. Analyzing these duplicates we find most of them represent challenging fine grained image classes (e.g. two similar dog breeds), or images where multiple annotations are appropriate. Additionally, in the Appendix G.2 we detail a second, harder to detect leak pattern we saw commonly in the training data with "near duplicates", images in the training set that are from the

| Model | Deduped? | Top1 | Top1 on Leaked | MLA | MLA on Leaked |
|-------|----------|------|----------------|-----|---------------|
| ResNet50 | - | 76.0% | 26.9% | 84.8% | 80.1% |
| ResNet50 | ✓ | 76.0% | 40.2% | 84.6% | 82.1% |
| ViT-3B | - | 89.5% | 42.5% | 97.8% | 93.4% |
| ViT-3B | ✓ | 89.4% | 45.1% | 97.4% | 94.0% |

Table 4: **Change in performance when removing leaked training examples**. We show both our ViT-3B and a ResNet50 for comparison, and report both Top1 accuracy and Multi-label Accuracy (MLA) on both the whole validation set and the leaked subset.

same photo-shoot or scene as a validation image, or are cropped or processed versions of a validation image. This phenomenon, with similar discovery methodology, has also been observed on CIFAR [3].

To understand the impact of this validation set leakage, we remove all the exact duplicates from the training set and retrain both a ResNet50 from scratch, and our JFT pretrained ViT-3B. Results are shown in Table 4. Unintuitively, when we remove all the leaked validation images from the training set and retrain, we see both Top1 Accuracy and Multi-label Accuracy (MLA) actually stay the same or decrease overall, despite all leaked training images having different labels than their leaked validation image counterpart. MLA accuracy both before and after deduplication are high, which leads us to believe that the training labels may be correct under multi-label evaluation. To verify this, we find 320 of the leaked validation images were in our 20k multi-label validation set, corresponding to 331 training images. Using these multi-labels, we find that 219/331 (66.2%) of these training images labels would have been correct under multi-label evaluation. Nevertheless, we do see an increase in both Top1 and MLA accuracies on the leaked subset of the data. Finally, it seems the ViT-3B model is less sensitive to leaked validation images, which may be a function of the fine tuning recipe we used (which was exactly the same as the fine-tuning recipe on the origianl non-deduped data).

## G.2  Near Duplicates

In addition to the 831 training examples that are exact duplicates of validation examples (all with different labels), there is a large collection of "near duplicates" in the Imagenet train set. "Near duplicates" are images that are either crops, augmentations, or resizes of validation images, or more unexpectedly, images from the same scene or photo shoot. These images are often visually different but semantically the same, and as a result are much harder to detect with traditional embedding distance thresholding based de-duplication. Nevertheless, these can also leak test set information, introduce label noise (if the labels are different than the validation set), or if many training examples are from the same scene, reduce the effective dataset size of that class.

While we cannot provide an estimate of the prevalence of this problem, we find it often while analyzing the K-Nearest Neighbors (in JFT embedding space) of validation images. We show some of these examples in Figures 14 and 15. While we just show dog and object examples here, we find this also happens with reasonable frequency for human related classes (especially activity related classes like soccer, basketball, parallel bars, etc). The existence of these duplicates raises an interesting question: How close can training examples be to your validation set before it becomes problematic?

## G.3  Neighbors of Spurious Correlations

While fine grained errors and OOV mistakes are often somewhat intuitive to a human, spurious correlations are harder to understand. To try to understand them in more detail, we look at the neighbors of several of the ViT-3B models major spurious correlations. In Figure 16 we show two such examples, but find this to be relatively common.

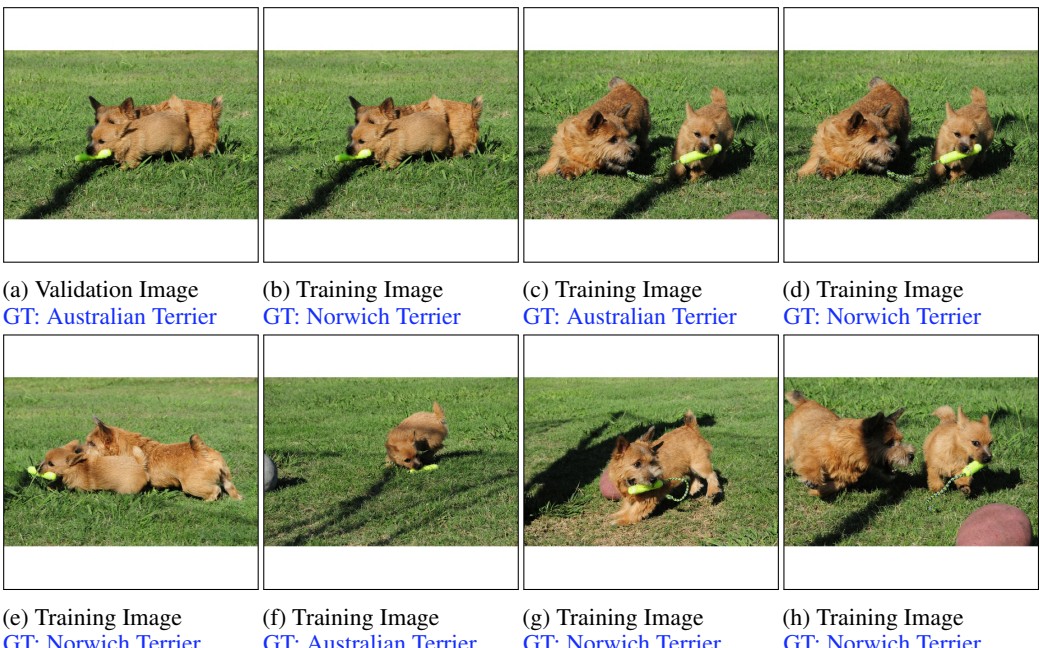

(a) Validation Image
GT: Australian Terrier

(b) Training Image
GT: Norwich Terrier

(c) Training Image
GT: Australian Terrier

(d) Training Image
GT: Norwich Terrier

(e) Training Image
GT: Norwich Terrier

(f) Training Image
GT: Australian Terrier

(g) Training Image
GT: Norwich Terrier

(h) Training Image
GT: Norwich Terrier

Figure 14: **Near Duplicates:** (a) We show a validation image of two dogs playing, labeled originally as an Australian Terrier. When looking at the K = 10 nearest neighbors, we find all 10 of them to be of the same two dogs with one of two labels, shown as images (b) through (h), including some training examples that are duplicates of each other. Because we only retrieve the 10 nearest neighbors, there could be even more than 10 images of this scene.

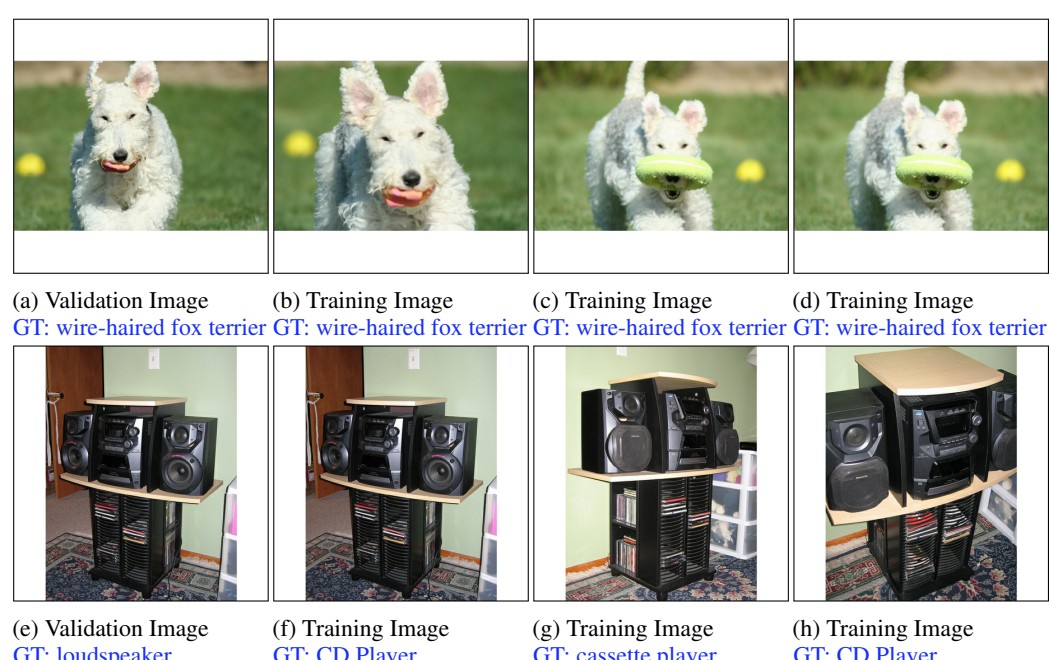

(a) Validation Image
GT: wire-haired fox terrier

(b) Training Image
GT: wire-haired fox terrier

(c) Training Image
GT: wire-haired fox terrier

(d) Training Image
GT: wire-haired fox terrier

(e) Validation Image
GT: loudspeaker

(f) Training Image
GT: CD Player

(g) Training Image
GT: cassette player

(h) Training Image
GT: CD Player

Figure 15: **Near Duplicates:** Top: (a) We show a wire-haired fox terrier with a cropped version of the validation image as a training image, and two more training images that are cropped versions of each other. We find 6/10 of the nearest neighbors are of the same dog. Bottom: We show a "loudspeaker" in the validation set with nearby images from the training set of the same speaker setup with labels of "CD Player" and "cassette player". We find 8/10 of its nearest neighbors are the same speaker setup, with several of the training images being exact duplicates of each other (or the validation set) with different labels.

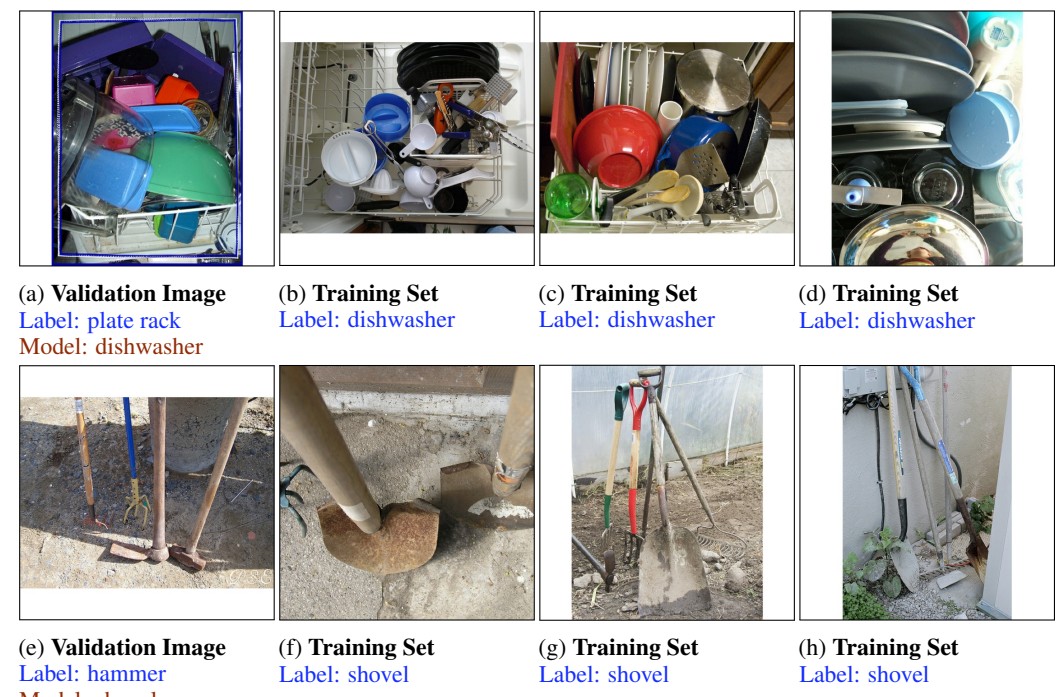

(a) **Validation Image**
Label: plate rack
Model: dishwasher

(b) **Training Set**
Label: dishwasher

(c) **Training Set**
Label: dishwasher

(d) **Training Set**
Label: dishwasher

(e) **Validation Image**
Label: hammer
Model: shovel

(f) **Training Set**
Label: shovel

(g) **Training Set**
Label: shovel

(h) **Training Set**
Label: shovel

Figure 16: **Neighbors of a Major Spurious Correlation.** We show an example of two major spurious correlations and a subset of their K=10 nearest neighbors (in JFT embedding space). For the dishwasher example (top), 8/10 of the nearest neighbors were pictures of cluttered dishes where the dishwasher machine was not in view. For the shovel example, we find all 10 nearest neighbors are shovels standing upright and outdoors. There are no other validation images of the hammer class with it standing upright or even outdoors.