# OpenReview forum: "When does dough become a bagel? Analyzing the remaining mistakes on ImageNet"
_NeurIPS.cc/2022/Conference — NeurIPS 2022 Accept_

### Official Review · Reviewer_NAFQ · 2022-06-24

**Rating:** 5
**Confidence:** 4
**Soundness:** 4 excellent
**Presentation:** 4 excellent
**Contribution:** 2 fair

**Summary:**

The paper analyses the remaining mistakes by the state of the art image classifiers on ImageNet. Several models with >95% on the multi-label accuracy (MLA) metric are considered and their mistakes are analysed. The paper discusses the remaining challenges in ImageNet and proposes a benchmark ImageNet-M that will quantify how well the future models would address those challenges.

**Questions:**

I think it's more natural to include the questions in the "strengths and weaknesses" section above.

**Limitations:**

I think it's more natural to include the limitations in the "strengths and weaknesses" section above.

**Strengths And Weaknesses:**

## Strengths

When evaluating a model, we often report only a number (e.g. classification accuracy) or a set of numbers (ImageNet variants). Then they miss out on the rich information about *what kind of mistakes* they make and *how severe* they are. Given that models are getting closer to 100% accuracy (in MLA), it is important to delve into the remaining errors of ImageNet to distinguish whether
- remaining errors are mostly due to labelling errors or beyond-the-common-sense difficulty; or
- they are reasonable errors that could be conquered.

The paper exactly does that. The conclusion is that there are both impossible and doable cases in the remaining errors. I believe this is interesting news for many who follow and contribute models on ImageNet.

The paper is quite interesting to read. The authors are often quite frank about the procedure and possible limitations there. This paper lets you think. I think it's a well-written paper.

## Weaknesses

ImageNet-M is a subset of the ImageNet validation set that combines the obvious (i.e. humans wouldn't really get confused - line 325)
errors made by four state-of-the-art models. The authors suggest this subset as another benchmark dataset for evaluating future close-to-perfect models.

I'm worried about the representativeness of ImageNet-M for the remaining challenges. Indeed, (roughly speaking) they are the intersection of the mistakes made by four state-of-the-art image classifiers. However, as the authors have observed as well, even the best models today make a different set of mistakes. I wonder if there is anything special about those 68 ImageNet-M samples. I'm worried that they are rather unlucky ones for the four particular classifiers.

Another way to formulate my worry is as follows: what can you conclude when model A gives 60/68 correct and model B gives 50/68 correct?
- Can you say model A is addressing the "remaining challenges" of ImageNet better than model B?
- Are you sure model A is not making more mistakes outside of ImageNet-M than model B?
- Isn't it sufficient to compare the multi-label accuracy (MLA) rather than ImageNet-M?
  - What additional benefit does ImageNet-M bring on top of MLA?
  - Will there be convincing use cases where ImageNet-M ranking would differ from MLA ranking?

One possible way to verify the representativeness of the 68 ImageNet-M images would be to do the following experiment: You prepare a sequence of K models M1, M2, M3, M4, M5, M6, ....., MK. Then, plot the size of ImageNet-M against the number of models used for generating ImageNet-M increases from 1 to K. I wish to see if
- the size of ImageNet-M converges to a non-zero number (ideally close to 68) as you include more models; or
- the size of ImageNet-M effective converges to zero as each model gets included.

If the former holds, then I would agree that ImageNet-M is indeed representative. If the latter holds, then ImageNet-M is rather a transient existence that results from considering the particular 4 models. I believe the trend from M1 up to M1,..,M4 (the four models for generating 68 images) would already tell us a lot. Could you share them?

The paper also seems to hint at the possibility of manually looking through the 68 samples to get more information about the model performance and mechanism than what you would get with numeric metrics (line 346). It would be great if the paper discusses how one could run such a qualitative analysis on the 68 samples - e.g. what kind of insights they could get and how they can improve the engineering based on that.

## Conclusion

The analysis of remaining errors for current models is interesting and useful (see "strengths"). However, I'm not too confident about ImageNet-M as a valid evaluation set for the remaining challenges for future models (see "weaknesses"). Unless the issues with ImageNet-M are successfully resolved in the rebuttal, I'd consider the paper's contribution to be insufficient for publication.

## After the rebuttal

I believe the authors have proved the minimal value for the ImageNet-M benchmark in their response. I would therefore change my score to 5 - borderline acceptance.

---

> ### Author Response · Authors · 2022-08-02
> **Response to Reviewer NAFQ**
>
> We thank the reviewer for their positive as well as constructive feedback on our work. In particular, we are happy you found our paper to be “interesting news for many who follow and contribute models on ImageNet” and “quite interesting to read…a well-written paper.”
>
> We respond to concerns about the utility of ImageNet-M in the joint response, and to individual questions and concerns inline below.
>
> `I wonder if there is anything special about those 68 ImageNet-M samples.`
>
> The special thing about the 68 images in ImageNet-M is that they were all “major errors” made by a majority of four very different SOTA models from recent years. They were the subset of errors remaining on ImageNet that we were confident as a team a human would get correct. The remaining errors were more borderline or minor, in many cases errors you may not bother fixing unless your application had very exacting requirements.
>
> ```
> Another way to formulate my worry is as follows: what can you conclude when model A gives 60/68 correct and model B gives 50/68 correct?
>   * Can you say model A is addressing the "remaining challenges" of ImageNet better than model B?
>   * Are you sure model A is not making more mistakes outside of ImageNet-M than model B?
> ```
> This is a great question, and in fact highlights the need to:
>
>   1. Evaluate ImageNet-M holistically with the other quantitative benchmarks like top-1 and MLA, rather than in isolation.
>   2. Analyze the *quality* of the mistakes made on this set.
>
> For example, let’s say that:
> * For model A:
>     * The 8 incorrect predictions are entirely non-sensical or pre-existing major mistakes.
>     * But gets slightly higher top-1 and MLA (say 0.2% higher on each than model B).
>
> * For model B:
>     * The 18 incorrect predictions are all novel predictions that would be deemed borderline.
>     * Gets slightly lower top-1 and MLA.
>
> One could argue that model B is actually the more useful model, because it never makes a major mistake on ImageNet-M, and on the broader set of top-1% and MLA is performing nearly on par.  Yes, it is conceivable that model B makes many major mistakes on an entirely novel set of MLA mistakes in that 0.2% gap, but given that 3 of 4 very different models all made mistakes on this common 68 set, we believe that is less likely.  For external evidence supporting this, see Section 3 of https://arxiv.org/pdf/2012.15483.pdf on dominance probabilities, which shows that slightly higher accurate models are unlikely to significantly ‘swap’ which examples they get wrong.
>
> ```
> Isn't it sufficient to compare the multi-label accuracy (MLA) rather than ImageNet-M?
>   * What additional benefit does ImageNet-M bring on top of MLA?
>   * Will there be convincing use cases where ImageNet-M ranking would differ from MLA ranking?
> ```
> MLA doesn’t capture the severity of the mistake, which is a critical thing you care about in the real world. ImageNet-M is a common set of MLA mistakes, and importantly it’s ones that are less ambiguous and that we are more confident in. We believe eventually a model will be able to get 100% on ImageNet-M, but don’t expect a model to be able to do that with MLA on the whole validation set.
>
> A model that improves upon MLA but does worse on ImageNet-M may just be getting better at solving ambiguous or borderline cases, and not improving solving examples we expect a model really should be able to get right. This looks a lot more like overfitting to a dataset.
>
> Finally, you can manually inspect ImageNet-M results easily due to its small size, and it is easier for the community to update.
>
> `Re: convincing use case.`
>
> Elucidating on the example in the shared response:   let’s say you have two models that both get around 91% on ImageNet top-1, and 98.5% on ImageNet-MLA.  One gets 55 of the 68 right, and the other gets 30 of the 68 right.  However, the predictions of the 55/68 correct model still makes 13 predictions that are major mistakes (e.g., they are the same mistakes other models have made), whereas the second model’s 38 incorrect predictions end up being novel predictions that would be categorized as borderline mistakes.  One could argue that the second model is actually more useful than the first, because it makes no major mistakes, and its novel predictions are in general closer to the right answer.  In applications like autonomous vehicles, if a model predicts a scooter for a bicycle, the set of actions taken are likely to be more similar / safe than if the scooter was mistaken for a shopping cart.
>
> Indeed, the point of ImageNet-M is not the quantitative result, but the qualitative one, which requires analyzing the quality of the mistake.  If the set were too big, researchers would find it hard to analyze, and if the set were too small, it might not be representative.  We’re not sure whether there’s a perfect size; we found 68 to be a reasonable number for someone to spend, without being trivially small (like < 20).

---

> > ### Author Response · Authors · 2022-08-02
> > **Response to Reviewer NAFQ (Cont.)**
> >
> > ```
> > One possible way to verify the representativeness of the 68 ImageNet-M images would be to do the following experiment: You prepare a sequence of K models M1, M2, M3, M4, M5, M6, ....., MK. Then, plot the size of ImageNet-M against the number of models used for generating ImageNet-M increases from 1 to K. I wish to see if
> >   * the size of ImageNet-M converges to a non-zero number (ideally close to 68) as you include more models; or
> >   * the size of ImageNet-M effective converges to zero as each model gets included.
> > ```
> > Here are the numbers:
> >
> > For ViT3B – we start with ~154 major mistakes.
> >
> > Vit3b_wrong & coca_wrong: 51
> > Vit3b_wrong & soups_wrong: 54
> > Vit3b_wrong & insta_wrong: 117
> >
> > Vit3b_wrong & soups_wrong & coca_wrong: 26
> > Vit3b_wrong & insta_wrong & coca_wrong: 36
> > Vit3b_wrong & insta_wrong & soups_wrong: 41
> >
> > all_wrong: 17
> >
> > We choose the slightly more relaxed ‘3 of 4 get them wrong’ criterion to get the 68 examples, rather than the ‘all of them wrong’ criterion that would yield only 17 examples.
> >
> > While we understand the concern about ‘representativeness’, we want to emphasize that the specific 68 examples themselves are not special in isolation.  They are merely a set that we’ve analyzed carefully where we believe 100% is achievable.  When people ask about whether ImageNet is done, people are in essence asking when models have solved the solvable examples.  ImageNet-M is an inexhaustive but careful attempt at measuring this.
> >
> > Moreover, for practical reasons, we tried to strike a balance between a slice that is too small to be general, and too large to be manually reviewed by the average researcher.  We expect most new SOTA models this year to get between 30-40 of them correct, so only 30-40 would need to be reviewed for potentially novel mistakes / categorization.
> >
> > We're happy to include this analysis in the Appendix to better contextualize ImageNet-M's creation.
> >
> > `It would be great if the paper discusses how one could run such a qualitative analysis on the 68 samples - e.g. what kind of insights they could get and how they can improve the engineering based on that.`
> >
> > This is a great question. We are happy to expand upon this in our paper. In general, we think researchers should use ImageNet-M for debugging and understanding the mistakes their models are making, and perhaps using it to identify common error patterns they may be able to address through targeted remediations. If a researcher evaluates their model on ImageNet-M and looks at their mistakes, they can:
> >
> > 1. Check that the mistakes are actually mistakes, and if not, potentially update ImageNet-M.
> > 2. See if the mistakes their model is making are actually major, for those that are they can identify what error category they think the mistake belongs in, and apply some appropriate remediation. If they are minor, that may be worth noting in a resulting paper for their method. A model that does not improve upon accuracy, but makes less (or no) “major” mistakes, if it can be communicated well, is also useful to the community (and practitioners!).
> > 3. As a researcher iterates on their model, they can keep evaluating on ImageNet-M to understand how their changes changed the distributions of mistakes a model is making.
> >
> > We hope that encouraging this style of analysis will lead to interesting follow on papers that look at specific remediations to remaining errors, or even provide a mistake remediation handbook, which would be impossible without a way to first understand and categories failure modes.

---

> > > ### Comment · Reviewer_NAFQ · 2022-08-07
> > > **Thanks for the response.**
> > >
> > > Thank you very much for the detailed response!
> > >
> > > I had concerns that
> > > 1. It's unclear how to use ImageNet-M to evaluate models (qualitatively and quantitatively) and decide the next steps based on the evaluation (debugging, model selection, model comparison, ...)
> > > 2. I wasn't entirely sure if ImageNet-M is sufficiently representative.
> > >
> > > The authors have argued that (if I'm understanding correctly)
> > > 1. ImageNet-M provides additional analysis on top of top-1 and MLA scores: how severe the errors actually are. The analysis can be run in a qualitative as well as quantitative fashion.
> > > 2. ImageNet-M is perhaps not "exhaustive", but model predictions tend to be similar (Section 3 of https://arxiv.org/pdf/2012.15483.pdf). One could also think of ImageNet-M as a necessary condition for a model to make no obvious errors.
> > >
> > > I think the response defends the essential value of ImageNet-M, though the contribution is still not very strong.
> > >
> > > ## Why the contribution is still not strong.
> > >
> > > It is not entirely convincing that there will be cases where the ImageNet-M scores will show drastically different behaviours than top-1 or multi-label accuracies. The examples discussed in the authors' response are hypothetical. Moreover, solving ImageNet-M does not ensure the elimination of all possible obvious errors and it may be possible that one could overfit to the particular samples in ImageNet-M.
> > >
> > > I find the authors' way of interpreting ImageNet-M results ambiguous. When model A is getting 30/68 correct and model B is getting 40/68 correct, while all the other accuracies (top-1 and MLA) are identical, we can argue that model B is better than model A. In that sense, the benchmark has a quantitative role - it ranks models and enables model selection. But at the same time, the authors suggest looking into the 68 samples to examine the models' qualitative behaviours. Will this activity bring further insights? - Such as model A is in fact better than model B in the example above because the remaining 28 mistakes by model B are much worse in quality than the 38 mistakes by model A. If this is the case, does this mean the simple ranking of models A and B based on the number of mistakes in ImageNet-M is not reliable? Alternatively, maybe ImageNet-M is intended purely as a dataset for qualitative analysis. In that case, we cannot compare models A and B quantitatively along with accuracies like top-1 and MLA. I find it hard to reconcile this double usage of ImageNet-M. I hope the final version of the paper contains more discussion about the actual use cases of the benchmark that involves both quantitative and qualitative treatment and explain how to interpret the results when the conclusions contradict each other.
> > >
> > > ## Corrected score: 5
> > >
> > > Reflecting the above, I would recommend borderline acceptance (score 5) for the paper.

---

### Official Review · Reviewer_VG4e · 2022-07-09

**Rating:** 6
**Confidence:** 4
**Soundness:** 3 good
**Presentation:** 2 fair
**Contribution:** 3 good

**Summary:**

The paper conducts extensive manual reviews on the mistakes made by a few top models. The ​mistakes are further annotated with severity and category. Two major findings are (1) about half of the mistakes are actually correct; (2) about 40% of the mistakes are major mistakes—predictions that humans find obviously wrong. The authors propose ImageNet-Major (ImageNet-M), an evaluation split based on 68 “major mistakes” images, to better benchmark models' progress on remaining mistakes.

**Questions:**

Questions
* I request the authors to answer if the “high error bar” problem (see weakness) would exist on ImageNet-M. If so, whether or not it can prevent ImageNet-M from serving as an evaluation dataset.
* Besides, could the authors clarify the distribution of these 68 images in terms of the mistake category (i.e., how many images are in the “Fine-grained” category and other categories)? Are the ImageNet-M the 68 images listed in Appendix E?

Minor suggestion
* Although it is good to present many details about the reviewing process, I feel it becomes very easy to get lost while reading the paper. The paper will be better presented if the authors add a figure (e.g., a flowchart) as an overview to summarize how the initial 676 mistakes are narrowed down to 378 mistakes and the final 68 major mistakes.


**Limitations:**

Yes, the authors adequately addressed them in Sec. 5.2.

**Strengths And Weaknesses:**

Strengths

* The paper studies an important problem to understand the remaining mistakes on ImageNet better.
* The human evaluations and the experiments are extensive.
* The manual review provides interesting findings (e.g., the model’s performance is underestimated due to label errors) and good insights (e.g., the categories of the mistakes).


Weaknesses

* My major concern is if the ImageNet-M can serve as an evaluation set for future models to benchmark. Due to its small size (i.e., only 68 images), the error bars (due to model selection or random seeds) could be very high.
* Some related works are not discussed. In terms of mistake analysis on ImageNet (L109-125), Salient ImageNet (Singla and Feizi 2022) annotated core and spurious features on ImageNet; Domino (Eyuboglu et al. 2022) designs a method to automatic detect systematical errors based on clustering. Regarding spurious correlation (L207-211) for the context, ImageNet-9 (Xiao et al. 2021) studies the influence of background on object recognition on ImageNet.

Singla, Sahil, and Soheil Feizi. 2022. “Salient ImageNet: How to Discover Spurious Features in Deep Learning?” In International Conference on Learning Representations. https://openreview.net/forum?id=XVPqLyNxSyh

Eyuboglu, Sabri, Maya Varma, Khaled Kamal Saab, Jean-Benoit Delbrouck, Christopher Lee-Messer, Jared Dunnmon, James Zou, and Christopher Re. 2022. “Domino: Discovering Systematic Errors with Cross-Modal Embeddings.” In International Conference on Learning Representations. https://openreview.net/forum?id=FPCMqjI0jXN.

Xiao, Kai Yuanqing, Logan Engstrom, Andrew Ilyas, and Aleksander Madry. 2021. “Noise or Signal: The Role of Image Backgrounds in Object Recognition.” In International Conference on Learning Representations. https://openreview.net/forum?id=gl3D-xY7wLq.

---

> ### Author Response · Authors · 2022-08-02
> **Response to Reviewer VG4e**
>
> We thank the reviewer for their positive and constructive feedback on our work. We respond to concerns about the utility of ImageNet-M in the joint response, and to individual questions and concerns inline below.
>
> `My major concern is if the ImageNet-M can serve as an evaluation set for future models to benchmark. Due to its small size (i.e., only 68 images), the error bars (due to model selection or random seeds) could be very high.`
>
> We appreciate the concern around error bars, which our clopper pearson intervals show on the figure in Section 5.1. As mentioned in our general response to reviewers, we want to emphasize that ImageNet-M is meant as less of a quantitative “model quality” score, and more of a “model capability measurement”.
>
> ImageNet-M is the remaining subset of mistakes these top performing models make that we A.) believe models should eventually be able to get 100% accuracy on, and B.) that we found existing SOTA models to make “major” mistakes on, mistakes we were sure a human would not make. A model that does really well on ImageNet-M, and slightly poorer on ImageNet Top1 might be interesting because it might be more “human like” in its behavior.
>
> Finally, as we state in our general response to reviewers, we view it’s small size as a feature, not a bug. We wanted the dataset to be small enough that a single researcher could reasonably look at all their model’s mistakes on this subset, and understand how their model improved, or identify what errors their model is still making, and the severity of them.
>
> `Some related works are not discussed. In terms of mistake analysis on ImageNet (L109-125), Salient ImageNet (Singla and Feizi 2022) annotated core and spurious features on ImageNet; Domino (Eyuboglu et al. 2022) designs a method to automatic detect systematical errors based on clustering. Regarding spurious correlation (L207-211) for the context, ImageNet-9 (Xiao et al. 2021) studies the influence of background on object recognition on ImageNet.`
>
> Thank you for these! We’re working on reading and understanding them so we can properly cite them in a final draft of this paper.
>
> `Besides, could the authors clarify the distribution of these 68 images in terms of the mistake category (i.e., how many images are in the “Fine-grained” category and other categories)? Are the ImageNet-M the 68 images listed in Appendix E?`
>
> This is a great suggestion, and we’re happy to add this to the paper and add all 68 images in ImageNet-M to the appendix with model predictions.

---

> > ### Comment · Reviewer_VG4e · 2022-08-07
> > **Concerns Addressed**
> >
> > The response addressed my concerns. I have raised my rating to "6: Weak Accept."
> >
> > But I still request the authors consider adding a flowchart as an overview to better illustrate how the initial 676 mistakes are narrowed down to the final 68 major mistakes.

---

### Official Review · Reviewer_c29F · 2022-07-16

**Rating:** 6
**Confidence:** 4
**Soundness:** 3 good
**Presentation:** 3 good
**Contribution:** 3 good

**Summary:**

The paper dives deeper on mistakes in ImageNet dataset by the latest models.  To help contextualize progress on ImageNet and provide a more meaningful evaluation for today’s state-of-the-art models, authors manually review and categorize the mistakes that a few top models make and provide insights into the long-tail of errors. Since ImageNet images contains multiple labels the paper focuses on the multi-label
subset evaluation of ImageNet, where today’s best models achieve upwards of 97% top-1 accuracy. The main contribution of the paper is what the analysis reveals:  nearly half of the supposed mistakes are not mistakes at all (significantly underestimating the performance
of these models). At the same time the models still make a significant number of mistakes (40%) that are obviously wrong to human
reviewers. To calibrate future progress on ImageNet, authors also provide an updated multi- label evaluation set.


**Questions:**

Have authors done similar analysis on another dataset to see how much the observations generalize? Even if it is a closed (proprietary) dataset?

Can we use this analysis to build better datasets or give guidelines to human reviewers or use it to quickly correct some other major datasets used by the community?

How do the authors think of this work translating to video classification where the accuracies are still on the lower side and the models are not as accurate?

**Limitations:**

1. Doing this evaluation on datasets that are larger like ImageNet22K (even if subsampled) would throw more light on the limitation of models. Right now, a lot of conclusions and observations is on the dataset rather than a particular set of models etc.

2. Also, going beyond classification into detection and segmentation would also have been a strong direction to think about.



**Strengths And Weaknesses:**

1. Thorough and useful analysis of mistakes made by current models on ImageNet
2. Significant clarity on the mistakes both qualitative (thanks to the supplementary material) and quantitative
3. A refined multi-label dataset for folks to compare in future
4. categorizing the mistakes into four: Fine-grained errors, Fine-grained with out-of-vocabulary, Spurious correlations, Non-prototypical labels.
5. Useful conclusions like 40% of errors are not errors, remaining errors have the four above categories, deduplication and nearest neighbor filtering from validation set.

---

> ### Author Response · Authors · 2022-08-02
> **Response to Reviewer c29F**
>
> We thank the reviewer for their positive and constructive feedback on our work. We respond to  individual questions and concerns inline below.
>
> `Have authors done similar analysis on another dataset to see how much the observations generalize? Even if it is a closed (proprietary) dataset?`
>
> We decided to focus on ImageNet because it’s the de-facto benchmark in vision. Additionally, because of the extensive time costs of expert review, we considered other datasets out of scope for this project. The fact that accuracy levels are so high on ImageNet makes manual review like this feasible.
>
> That said, our ImageNetV2 results show that the general issues hold across a dataset collected independently of ImageNet, which suggests these behaviors are likely to occur whenever the set of classes is overlapping enough to be multi-label. It would not be surprising if these results held on other datasets with similar label set properties.
>
> `Can we use this analysis to build better datasets or give guidelines to human reviewers or use it to quickly correct some other major datasets used by the community?`
>
> We absolutely believe that the community can use our process to build better evaluation datasets. One of the major lessons of this work is that evaluation datasets need to be continually updated throughout their lifecycle as models get better.
>
> In Section 4.2 we state: “These proportions suggest that large models are frequently uncovering new correct multi-labels, suggesting that mistake analysis and label correction needs to be part of the lifecycle and of benchmark development of long-tailed errors to properly assess performance as a benchmark saturates.”
>
> Additionally, as we state in the paper, benchmark designers should be more careful in designing their tasks such that humans are plausibly able to label things correctly when given sufficient time, and that they should think not just of the initial state of the benchmark, but what it means to “solve it”. Much hand-wringing in the community around ImageNet lately is precisely because the benchmarks aren't clear about what to do as models approach perfection.
>
> `How do the authors think of this work translating to video classification where the accuracies are still on the lower side and the models are not as accurate?`
>
> While video benchmarks seem less close to saturation of performance, we’ve seen similar types of issues with video classification, in that a scene might have multiple plausible labels depending on the dataset, and yet we’re still evaluating without multi-labels typically.  We expect similar types of issues to arise as performance on these benchmarks approaches saturation too.
>
> `Also, going beyond classification into detection and segmentation would also have been a strong direction to think about.`
>
> This is a great direction for future work. One thing worth noting is that some of the issues we see in classification actually go away in localization tasks like detection and segmentation, because objects or things are separately labeled with their respective pixels. This removes some of multi-label ambiguity plaguing ImageNet, but we expect some issues like fine grained classification to still exist given a sufficiently fine grained label list, and for things like spurious correlations to still exist.

---

### Author Response · Authors · 2022-08-02
**General Response to Reviewers**

We deeply thank the reviewers for their time and thoughtful feedback on our work. We are glad to hear that Reviewer c29F found our analysis “thorough and useful”, Reviewer VG4e found our human evaluations “extensive” and addressing an “important problem” with “interesting findings…and good insights”, and Reviewer NAFQ found our paper “well-written” and interesting and thought-provoking to read.

We also hear your concerns regarding the utility of our proposed dataset ImageNet-M. We have clarified the purpose of this dataset (how we hope the community will use and benefit from it in our response below, and will revise it accordingly in our paper. We also address reviewer-specific questions in our individual response to each reviewer. Please feel free to follow up with any additional questions.

**ImageNet-M**

Reviewer VG4e and Reviewer NAFQ shared similar concerns about the size and representativeness of ImageNet-M.

ImageNet-M is the remaining subset of mistakes these top performing models make that we A.) are confident a model should eventually be able to reach 100% accuracy on, and B.) found existing SOTA models to make “major” mistakes on, mistakes we were quite sure a human would not make.

As the Clopper-Pearson intervals show, ImageNet-M is best suited as a qualitative analysis slice of ImageNet multi-label evaluation, and should be reported and analyzed as context for both top-1 and MLA quantitative numbers. We will work to emphasize this more in the paper.

Our goal here is not to add yet another context-less quantitative number for researchers to put in their paper, but instead provide a small slice of examples (where we have confidence in the labels) that is reasonable for a single researcher to manually review to find out *how their model improved*, instead of *how much their model improved*. We hope that authors of future papers use ImageNet-M as a barometer of the quality of the mistakes their model makes, and as a tool to understand and fix model failures.

ImageNet-M is unique among benchmarks and evaluation splits of ImageNet in that we believe it is possible to get 100% on it, and research that aims in solving the remaining long tail of a benchmark (e.g. active learning) will benefit greatly from ImageNet-M. It provides researchers a trustworthy way to ask “How well does our technique help comprehensively solve major errors made even by todays SOTA models?”.

Finally, when considering ImageNet-M as a qualitative tool for analyzing model predictions, we find its small size to be a feature rather than a bug. We wanted the dataset to be small enough that a single researcher could reasonably look at all their model’s mistakes on this subset. For example, as one of the reviewers questioned, if model A gets 50/68 right, a careful analysis of the remaining 18 mistakes might show that the errors are all borderline, whereas the hypothetical  60/68 model might still make 8 very egregious mistakes. That may be an important difference, especially if the two models are roughly comparable on ImageNet Top-1 and MLA. This way, a qualitative ImageNet-M analysis adds nuance to existing quantitative numbers.

---

### Meta-Review · Area_Chair_6GPt · 2022-08-26

**Recommendation:** Accept
**Confidence:** Certain

**Metareview:**

All reviewers are positive about this paper, leaning toward accept. The AC does not find sufficient grounds to overrule the consensus.

**Award:**

No

---

### Decision · Program_Chairs · 2022-09-14

Accept